# Learning Video Generation for Robotic Manipulation with Collaborative Trajectory Control

**Xiao Fu**[1]   **Xintao Wang**[2✉]   **Xian Liu**[1]   **Jianhong Bai**[3]   **Runsen Xu**[1]
**Pengfei Wan**[2]   **Di Zhang**[2]   **Dahua Lin**[1✉]
[1]The Chinese University of Hong Kong   [2]Kuaishou Technology   [3]Zhejiang University

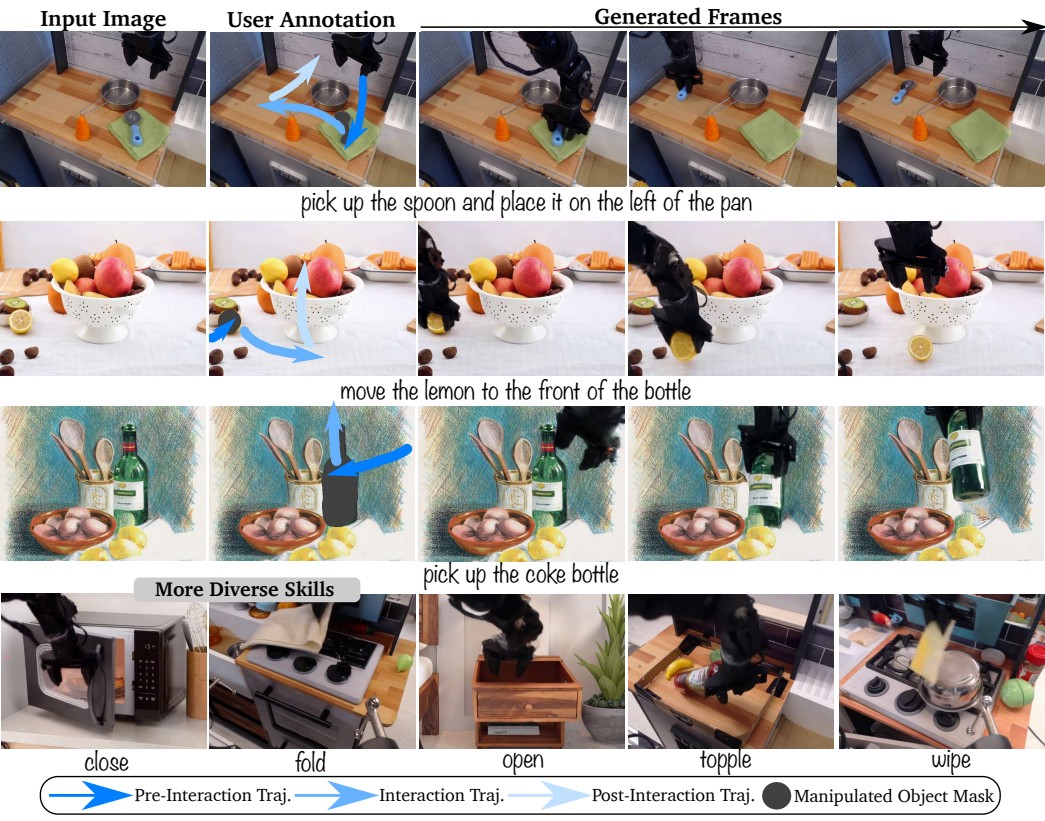

**Figure 1: RoboMaster** synthesizes realistic robotic manipulation video given an initial frame, a prompt, a user-defined object mask, and a collaborative trajectory describing the motion of both robotic arm and manipulated object in decomposed interaction phases. It supports diverse manipulation skills and can generalize to in-the-wild scenarios. Please check more on our website.

## Abstract

Recent advances in video diffusion models shows promise for generating robotic decision-making data, with trajectory conditions further enabling fine-grained control. However, existing methods primarily focus on *individual object motion* and struggle to capture *multi-object interaction* crucial in complex manipulation. This limitation arises from entangled features in overlapping regions, leading to degraded visual fidelity. To address this, we present *RoboMaster*, a novel framework that models inter-object dynamics via a collaborative trajectory formulation. Unlike prior methods that decompose objects, our core is to decompose the interaction process into three sub-stages: pre-interaction, interaction, and post-interaction, and models each phase using the dominant object, specifically the robotic arm in

✉: Corresponding Authors.

the pre- and post-interaction phases and the manipulated object during interaction. This design effectively alleviates the multi-object feature fusion issue in prior work. To further ensure subject semantic consistency across the video, we incorporate appearance- and shape-aware latent representations for objects. Extensive experiments on the challenging Bridge dataset, as well as RLBench and SIMPLER benchmarks, demonstrate that our method establishs new state-of-the-art performance in trajectory-controlled video generation for robotic manipulation. Project Page: `https://fuxiao0719.github.io/projects/robomaster/`

# 1 INTRODUCTION

Embodied AI has achieved remarkable progress in recent years (Brohan et al., 2022; 2023; Cheang et al., 2024; Bjorck et al., 2025; Lynch et al., 2023; O'Neill et al., 2024; Li et al., 2023), holding promise to replace human labor in performing diverse tasks. Scalable robot learning plays a crucial role in empowering embodied intelligent agents to fulfill diverse and generalizable skills in unseen environments. However, a major bottleneck remains: *data scarcity* (Yu et al., 2024a; Liu et al., 2024b). Collecting large-scale data using real robots is costly and requires human supervision to ensure safety.

Recently, video generation (Peebles & Xie, 2023; Yang et al., 2024b; Agarwal et al., 2025; Wang et al., 2025; Kong et al., 2024; Bai et al., 2025) has emerged as a promising approach for simulating realistic environments, offering visually plausible content that closely resembles the real world. Leveraging this, several works have explored generating robotic decision-making data from multimodal inputs, e.g., instruction (Du et al., 2023; Yang et al., 2023; Team, 2024; Ko et al., 2023), sketch (Zhou et al., 2024), and trajectory (Zhu et al., 2024; Team, 2024). Among these, trajectory-conditioned generation enables fine-grained control over robot planning by structuring the motions of both the robotic arm and the manipulated object. However, previous works, such as Tora (Zhang et al., 2025) and DragAnything (Wu et al., 2024), simply focus on driving individual object motion with separate trajectories (see Tab. 1). This design leads to feature entanglement in overlapping regions during interaction (highlighted by the red box in Fig. 2), which impairs the model's ability to capture physically plausible interactions and impairs visual fidelity. In robot learning, high-quality demonstrations from video world models (Ali et al., 2025; Agarwal et al., 2025) can derive executable action labels via inverse dynamics models for downstream action planning. However, if the synthesized video fails to accurately capture the interaction phase, the inverse dynamics model may extract unreliable actions, potential limiting the effectiveness of the learned robotic policy.

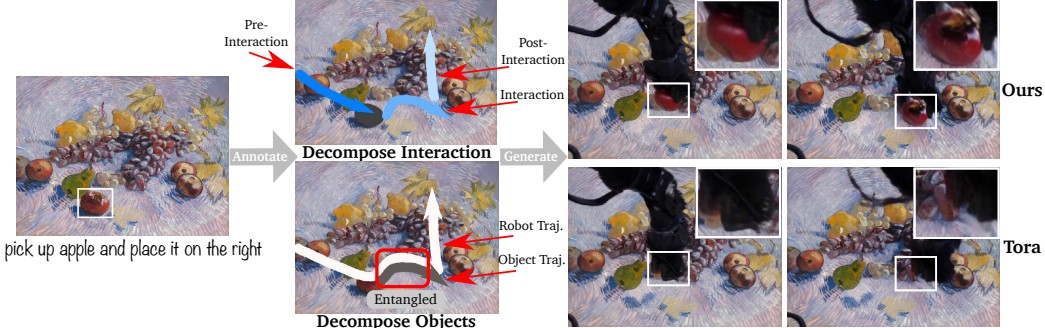

Figure 2: **Collaborative Trajectory (Ours) vs Separated Trajectories (Previous, e.g.Tora).** Unlike Tora (Zhang et al., 2025) that *decomposes objects* and model the motion of robot arm and manipulated object separately, we *decompose the interaction phase* and unify their joint motions into a single collaborative trajectory with fine-grained object awareness. This integration alleviates the feature fusion issue in overlapping regions (see the missing apple in Tora), and improves visual quality.

To address these limitations, we propose *RoboMaster*, which models robotic manipulation with a novel collaborative trajectory. Unlike Tora, which *decomposes multi-object motion* using separate trajectories, RoboMaster captures interactive dynamics within a unified trajectory representation. Specifically, we *decompose the interaction process* into three sub-phases: pre-interaction, interaction, and post-interaction. Each phase is guided by the dominant agent, namely the robotic arm in the pre-

and post-interaction phases, and the manipulated object during interaction. This design is motivated by the observation that the robotic arm initiates and concludes the motion, while the object remains largely static; during interaction, the object's motion reflects the physical response to manipulation, implicitly synchronizing with the robotic arm's trajectory. By explicitly modeling the driving subjects and their features across interaction phases, RoboMaster mitigates the feature entanglement issues and facilitates learning plausible interactions, rather than strictly following trajectory by compromising interaction fidelity. Furthermore, to ensure semantic consistency of the manipulated object throughout the video sequence, we leverage the user-defined object mask to sample encoded RGB latents. These latents are then associated with the object's shape to construct circular volumetric representation, preserving both appearance and shape across frames.

Table 1: **Comparison of Ours with Previous Trajectory-Controlled Methods.**

| | Interaction Granularity | | Object Awareness | | |
|---|---|---|---|---|---|
| | Trajectory | Decomposed? | Format | Appearance | Shape |
| IRAsim (Zhu et al., 2024) | Single (Isolated) | ✗ | N/A | ✗ | ✗ |
| DragAnything (Wu et al., 2024) | Multiple (Isolated) | ✗ | Mask | ✓ | ✓ |
| Tora (Zhang et al., 2025) | Multiple (Isolated) | ✗ | Point | ✗ | ✗ |
| **RoboMaster (Ours)** | Single (Collaborative) | ✓ | Mask | ✓ | ✓ |

Beyond improved manipulation modeling, our design also facilitates user interaction: 1) user can easily annotate the interaction phase instead of full arm–object trajectories, simplifying trajectory correction under annotation errors 2) user can flexibly specify object regions with a brush tool. Our experiments demonstrate that RoboMaster remains robust even with imprecise user input.

We conduct extensive experiments on the challenging Bridge dataset (Walke et al., 2023) and demonstrate that RoboMaster outperforms prior trajectory-controlled video generation methods in both visual quality and trajectory accuracy. Further evaluations on RLBench (James et al., 2020) and SIMPLER (Li et al., 2024) benchmarks validate its effectiveness for robotic action planning. Our contributions are as follows:

1) We introduce a collaborative trajectory framework that models robotic manipulation by decomposing the interaction phase into sub-phases, enabling video generation as an interactive simulator for high-quality robotic data. We also contribute a high-quality 21k video–trajectory dataset from Bridge.

2) Our design combines collaborative trajectories with mask-based object embeddings, allowing for more intuitive user annotation and significantly enhancing user interactivity.

3) Extensive experiments show that our approach achieves state-of-the-art results on both visual and robotic action planning benchmarks, outperforming existing trajectory-conditioned methods.

## 2 RELATED WORK

**Trajectory-Controlled Video Generation for Object Movement.** Early works (Wang et al., 2024d; Ma et al., 2024; Mou et al., 2024; Yin et al., 2023; Zhang et al., 2025) leverage point-based control to enhance adaptability and user interactivity. Subsequent methods adopt mask-based representations (including bounding boxes) (Yang et al., 2024a; Qiu et al., 2024; Wang et al., 2024b; Wu et al., 2024; Dai et al., 2023) or optical flow (Geng et al., 2024a; Shi et al., 2024) to improve robustness over point-based alternatives. Beyond 2D representations, 3DTrajMaster (Fu et al., 2025) and ObjCtrl-2.5 (Wang et al., 2024c) model object motion using 6-DoF trajectories, while LeviTor (Wang et al., 2024a) incorporates depth to enable 3D-aware object manipulation. However, existing works overlook interaction scenarios and treat object motion as independently controlled, which degrades visual quality in overlapping regions. In contrast, RoboMaster introduces collaborative trajectory control, unifying interactive features and decomposed trajectories to model interaction effectively.

**Video Generation as World Simulator for Robotic Manipulation.** Scalable robot learning (Brohan et al., 2022; 2023; Cheang et al., 2024; Bjorck et al., 2025; Lynch et al., 2023) relies heavily on large-scale realistic data, but collecting real-world robot trajectories from human demonstrations remains time-consuming and labor-intensive, limiting public accessibility. To address this, generative video models (Wu et al., 2023; Agarwal et al., 2025) offer a cost-effective alternative for synthesizing realistic data for policy learning. UniPi (Du et al., 2023) and AVDC (Ko et al., 2023) frame robot

planning as text-to-video generation, with AVDC further incorporating inverse dynamic estimation via a pretrained flow network. UniSim (Yang et al., 2023) learns a unified real-world simulator with diverse conditions (text and control inputs). TesserAct (Zhen et al., 2025) learns 4D robotic manipulation by additional modeling video depth&normal. IRASim (Zhu et al., 2024) also employs trajectory-conditioned video generation but only models robot arm motion. In contrast, our method jointly models both robot and object trajectories with fine-grained object awareness, enabling higher visual quality and greater user interactivity in unseen scenarios.

## 3 METHOD

Our goal is to enable fine-grained and user-friendly control in image-to-video generation for robotic manipulation. To this end, we present *RoboMaster* (see Fig. 3), a framework built upon a *collaborative trajectory mechanism*. We begin by reviewing the prior trajectory control paradigm and outlining our task formulation (Sec. 3.1). We then introduce the key components required for control: (1) object embeddings that encode appearance and shape to maintain identity consistency (Sec. 3.2), and (2) collaborative trajectory that models the interactive dynamics (Sec. 3.3). These are integrated via a motion injector (Sec. 3.4) that effectively guides motion generation.

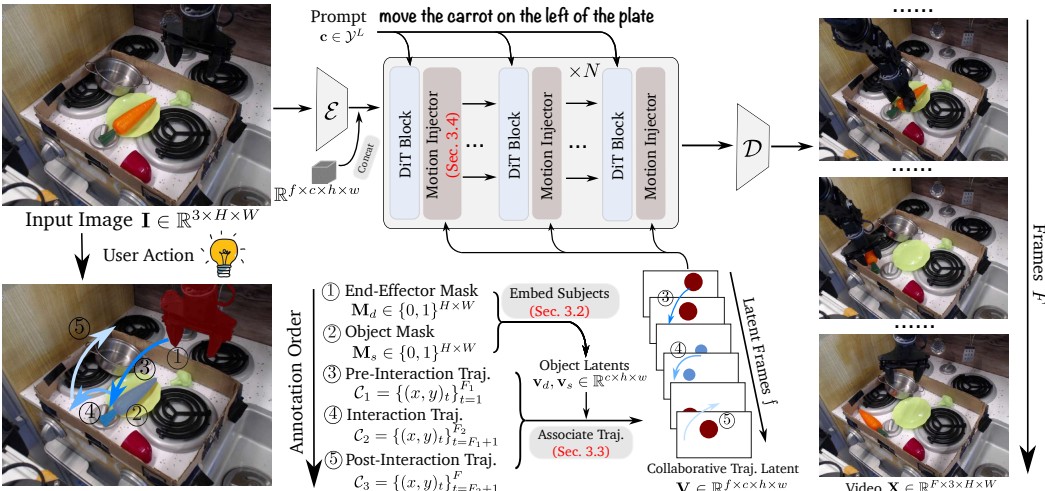

Figure 3: **RoboMaster Framework.** Given an input image $\mathbf{I}$ and a prompt $\mathbf{c}$, it generates a desired robotic manipulation video $\mathbf{X}$ with the collaborative trajectory design. Specifically, it first encodes the object masks, including robotic arm $\mathbf{M}_d$ and submissive object $\mathbf{M}_s$ (acquired either from 1) Grounded-SAM Ren et al. (2024) or 2) user-defined brush mask) with the awareness of appearance and shape to obtain $\mathbf{v}_d, \mathbf{v}_s$ for maintaining identity consistency in the video. To precisely model the manipulation process, the controlled trajectory $\mathcal{C}$ is decomposed into sub-interaction phases: pre-interaction $\mathcal{C}_1$, interaction $\mathcal{C}_2$, and post-interaction $\mathcal{C}_3$, associating each phase with object-specific latents $\mathbf{v}_d$, $\mathbf{v}_s$, and $\mathbf{v}_d$, respectively. The collaborative trajectory latent $\mathbf{V}$ is then injected into plug-and-play motion injectors, enabling the reasoning of video dynamics during generation.

### 3.1 PRELIMINARY: VIDEO DiTs WITH DECENTRALIZED TRAJECTORY CONTROL

Video diffusion transformers (DiTs) Peebles & Xie (2023); Lin et al. (2024); Yang et al. (2024b); Agarwal et al. (2025); Wang et al. (2025); Kong et al. (2024) with trajectory condition $\mathcal{C}$ learns the conditional distribution $p(\mathbf{x} \mid \{\mathcal{C}_n\}_{n=1}^{N})$ of the compressed video data $\mathbf{x} = \text{patchify}(\mathcal{E}(\mathbf{X}))$, where N is the trajectory number, $\mathcal{E}(\cdot)$ is a 3D VAE encoder and $\mathbf{X} \in \mathbb{R}^{F \times 3 \times H \times W}$ is clean video. It involves a forward process q to progressively inject noise $\boldsymbol{\epsilon}$ on $\mathbf{x}_0$ to the desired Gaussian distribution in a Markov chain: $\{\mathbf{x}_t, t \in (1, T) \mid \mathbf{x}_t = \alpha_t \mathbf{x}_0 + \sigma_t \boldsymbol{\epsilon}, \boldsymbol{\epsilon} \sim \mathcal{N}(\mathbf{0}, \mathbf{I})\}$, and a reverse process $p_{\boldsymbol{\theta}}$ to remove noise via a noise estimator $\hat{\boldsymbol{\epsilon}}_{\boldsymbol{\theta}}$, trained by minimizing:

$$\min_{\boldsymbol{\theta}} \mathbb{E}_{t \sim \mathcal{U}(0,1), \boldsymbol{\epsilon} \sim \mathcal{N}(\mathbf{0}, \mathbf{I})} \left[ \left\| \hat{\boldsymbol{\epsilon}}_{\boldsymbol{\theta}} \left( \boldsymbol{x}_t, t, \{\mathcal{C}_n\}_{n=1}^{N} \right) - \boldsymbol{\epsilon} \right\|_2^2 \right] \tag{1}$$

**Task Formulation** Given an initial frame $\mathbf{I}$ containing interaction subjects, a dominant subject $\mathbf{o}_d$ and a submissive subject $\mathbf{o}_s$, along with user-defined text prompt $\mathbf{c}$, binary object masks $\mathbf{M}_d$ [1] and $\mathbf{M}_s$ (where $\mathbf{M} \in \{0, 1\}^{H \times W}$), and a collaborative trajectory $\mathcal{C} = \{(x, y)_t\}_{t=1}^F$, our objective is to synthesize a plausible manipulation video $\mathbf{X}$. The trajectory $\mathcal{C}$ is structured into three temporal phases: pre-interaction $\mathcal{C}_1 = \{(x, y)_t\}_{t=1}^{F_1}$, interaction $\mathcal{C}_2 = \{(x, y)_t\}_{t=F_1+1}^{F_2}$, and post-interaction $\mathcal{C}_3 = \{(x, y)_t\}_{t=F_2+1}^F$. We define the general formulation $f_{\boldsymbol{\theta}}(\cdot)$ of the generative model as

$$f_{\boldsymbol{\theta}}(\cdot) : \mathbf{I} \in \mathbb{R}^{3 \times H \times W}, \mathbf{c} \in \mathcal{Y}^L, \mathbf{M}_d, \mathbf{M}_s \in \{0, 1\}^{H \times W}, \mathcal{C} \in \{(x, y)_t\}_{t=1}^F \to \mathbf{X} \in \mathbb{R}^{F \times 3 \times H \times W} \tag{2}$$

where $\mathcal{Y}$ is the alphabet, $L$ is the token length, and $\mathbf{X} \approx \mathcal{D}(\text{unpat\^{c}hify}(\mathbf{x}_0))$.

## 3.2 SUBJECT REPRESENTATION VIA COUPLED APPEARANCE AND SHAPE EMBEDDING

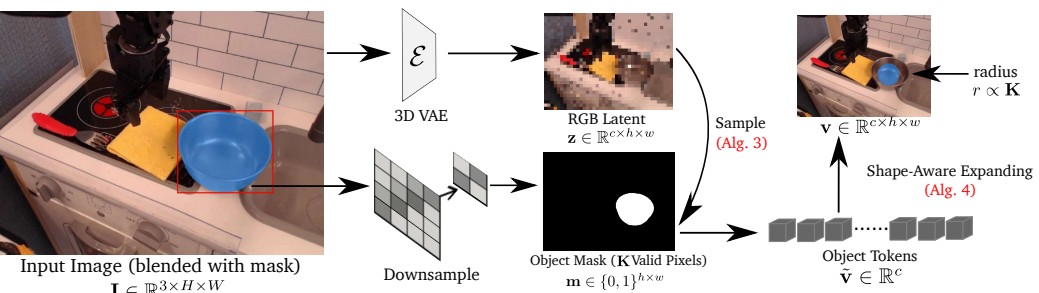

Figure 4: **Subject Embedding Illustration.** The object mask $\mathbf{M}$ is interpolated to align with the encoded RGB latents $\mathbf{z}$. Then it samples $\mathbf{z}$ with valid pixels and applies an average pooling operator to generate the embedding $\tilde{\mathbf{v}}$. To enhance spatial awareness, it expands the object token by a radius $r$, which is proportional to the area of the valid mask region, and obtains the circular volume $\mathbf{v}$.

As shown in Fig. 4, the initial frame $\mathbf{I}$ is first projected into latent features $\mathbf{z}$ via the VAE encoder $\mathcal{E}(\cdot) : \mathbf{I} \in \mathbb{R}^{3 \times H \times W} \to \mathbf{z} \in \mathbb{R}^{c \times h \times w}$ with spatial compression factors $c_s$. The object masks are subsequently downsampled using an interpolation operator $\mathcal{F}_d(\cdot) : \mathbf{M}_d, \mathbf{M}_s \in \{0, 1\}^{H \times W} \to \mathbf{m}_d, \mathbf{m}_s \in \{0, 1\}^{h \times w}$ to match the spatial resolution of the latent feature map. We then extract the latent subject features by applying the corresponding masks to $\mathbf{z}$, followed by pooling operator, resulting in $\tilde{\mathbf{v}}_d, \tilde{\mathbf{v}}_s \in \mathbb{R}^c$, defined as:

$$\tilde{\mathbf{v}}_{d,s}[i] = \frac{1}{\sum_{i=1}^h \sum_{j=1}^w \mathbf{m}_{d,s}[i, j]} \sum_{i=1}^h \sum_{j=1}^w \tilde{\mathbf{z}}_{d,s}[i, x, y] \quad \text{for} \quad i = 0, 1, ..., c \tag{3}$$

$$\tilde{\mathbf{z}}_{d,s}[i, x, y] = \mathbf{z}[i, x, y] \quad \text{if} \quad \mathbf{m}_{d,s}[x, y] = 1 \quad \text{otherwise} \quad \tilde{\mathbf{z}}_{d,s}[c, x, y] = 0$$

At each timestep $t$ within the latent video length $f$, which is the temporal-compressed length of $F$, we represent the subjects as *circular volume* $\mathbf{v}_d, \mathbf{v}_s \in \mathbb{R}^{c \times h \times w}$, centered at the trajectory point $(x, y)_t$ and confined to the corresponding valid mask region. This volume is constructed as:

$$\mathbf{v}_{d,s}[i, j, k] = \tilde{\mathbf{v}}_{d,s}[i] \quad \text{if} \ (j - x)^2 + (k - y)^2 <= r_{d,s}^2 \quad \text{otherwise} \quad \mathbf{v}_{d,s}[i, j, k] = 0 \quad \forall i, j, k \tag{4}$$

where the radius $r_{d,s}$ is proportional to the mask area, i.e., $r \propto \sum_{i=1}^h \sum_{j=1}^w \mathbf{m}[i, j]$. Incorporating both object appearance and spatial shape into this latent representation accelerates training convergence and improves identity consistency across subsequent frames in the video sequence.

## 3.3 COLLABORATIVE TRAJECTORY REPRESENTATION

Decentralized modeling of multiple trajectories, represented as $p_{\boldsymbol{\theta}}(\mathbf{x} \mid \mathbf{I}, \mathbf{c}, \{\mathcal{C}_n\}_{n=1}^N)$, is appropriate for scenarios where objects follow independent motion patterns without mutual intervention. However, when applied to interactive scenarios, e.g., picking up or moving objects, it exhibits several

---

[1]We omit the robotic arm mask in Fig. 1 as 1) for better illustration 2) user is not required to additionally input it and only need to use the pre-defined mask instead.

limitations: 1) *Feature overlap*: Interaction phase dominates the overall motion, and it introduces feature ambiguity in such overlapping regions as models are primarily trained on independently moving objects, leading to degraded synthesized quality. 2) *Trajectory precision during interaction*: Accurately specifying the trajectory of the dominant subject $\mathbf{o}_d$ during the interaction phase is challenging. Users may find it difficult to define precise temporal boundaries (e.g., start and end timestamps) and relative spatial positioning with respect to the submissive object $\mathbf{o}_s$.

To address these limitations, we propose learning a unified distribution $p_{\boldsymbol{\theta}}(\mathbf{x} \mid \mathbf{I}, \mathbf{c}, \mathbf{v}_d, \mathbf{v}_s, \mathcal{C})$ with collaborative trajectory $\mathcal{C}$, which is further temporally decomposed into three subsets:

**Pre-&Post-Interaction** During these phases, the dominant subject $\mathbf{o}_d$ serves as the sole moving agent, while the submissive subject $\mathbf{o}_s$ remains static or exhibits minor motion due to inertia. Accordingly, we leverage the dominant subject's trajectory $\mathcal{C}_1 = \{(x_d, y_d)_t\}_{t=1}^{F_1}$, $\mathcal{C}_3 = \{(x_d, y_d)_t\}_{t=F_2+1}^{F}$ along with its circular volume $\mathbf{v}_d$ to model the distribution $p_{\boldsymbol{\theta}}(\mathbf{x}_1 \mid \mathbf{I}, \mathbf{c}, \mathbf{v}_d, \mathcal{C}_1)$ and $p_{\boldsymbol{\theta}}(\mathbf{x}_3 \mid \mathbf{I}, \mathbf{c}, \mathbf{v}_d, \mathcal{C}_3)^2$.

**Interaction** At this stage, the interactive agents $\mathbf{o}_d$ and $\mathbf{o}_s$ collaborate to carry out the instruction $\mathbf{c}$. We incorporate submissive subject's trajectory $\mathcal{C}_2 = \{(x_s, y_s)_t\}_{t=F_1+1}^{F_2}$ and its corresponding circular feature $\mathbf{v}_s$ to model the conditional distribution $p_{\boldsymbol{\theta}}(\mathbf{x}_2 \mid \mathbf{I}, \mathbf{c}, \mathbf{v}_s, \mathcal{C}_2)$. Our intuition is twofold: 1) the motion of the submissive subject can implicitly guide the dominant subject, owing to the typically constrained relative dynamics between interacting entities during this phase 2) temporal variations in the feature representation (i.e., $\mathbf{v}_d \rightarrow \mathbf{v}_s \rightarrow \mathbf{v}_d$) can provide valuable cues for modeling behavioral changes (sole object movement $\rightarrow$ interactive objects movement) over time.

**Causal Representation.** Given the causal nature of the 3D VAE encoder $\mathcal{E}(\cdot)$, we incorporate latent feature map from previous frames into subsequent ones to enhance smoother transitions. Specifically, at each timestep $t$, latent feature map from timestep $t-1$ is propagated forward, and the current object feature ($\mathbf{v}_d$ or $\mathbf{v}_s$) is overwritten onto it. Consequently, the interaction and post-interaction distributions are updated as $p_{\boldsymbol{\theta}}(\mathbf{x}_2 \mid \mathbf{I}, \mathbf{c}, \mathbf{v}_d, \mathbf{v}_s, \mathcal{C}_1, \mathcal{C}_2)$ and $p_{\boldsymbol{\theta}}(\mathbf{x}_3 \mid \mathbf{I}, \mathbf{c}, \mathbf{v}_d, \mathbf{v}_s, \mathcal{C}_1, \mathcal{C}_2, \mathcal{C}_3)$

In general, our collaborative design factorizes the vanilla distribution $p_{\boldsymbol{\theta}}(\mathbf{x} \mid \mathbf{I}, \mathbf{c}, \mathcal{C}_s, \mathcal{C}_d)$ into multiple object-aware sub-distributions, thereby alleviating feature confusion and improving interaction:

$$\underbrace{p_{\boldsymbol{\theta}}(\mathbf{x}_1 \mid \mathbf{I}, \mathbf{c}, \mathbf{v}_d, \mathcal{C}_1)}_{\text{pre-interaction}} \underbrace{p_{\boldsymbol{\theta}}(\mathbf{x}_2 \mid \mathbf{I}, \mathbf{c}, \mathbf{v}_d, \mathbf{v}_s, \mathcal{C}_1, \mathcal{C}_2)}_{\text{interaction}} \underbrace{p_{\boldsymbol{\theta}}(\mathbf{x}_3 \mid \mathbf{I}, \mathbf{c}, \mathbf{v}_d, \mathbf{v}_s, \mathcal{C}_1, \mathcal{C}_2, \mathcal{C}_3)}_{\text{post-interaction}} \tag{5}$$

**User Interaction** Our design offers several key advantages for user-friendly access to generalizable experiments: 1) *Robustness in object extraction*: Due to mask-based representation, users can flexibly specify interaction object using a simple brush tool. Our experiments show that object identity remains well-preserved, even with a coarse input brush-based mask, in contrast to a complete one generated with SAM Ravi et al. (2024). 2) *Flexibility in input trajectory*: instead of requiring two full-length trajectories, users can define decomposed sub-trajectories within a single motion path. This not only simplifies the input process but also enhances adaptability for iterative refinement.

### 3.4 Motion Injection Module

The collaborative trajectory latent $\mathbf{V} \in \mathbb{R}^{f \times c \times h \times w}$, which associates $\mathbf{v}_d, \mathbf{v}_s$ with latent frame length $f$, is patchified and sequentially encoded by a zero-initialized 2D spatial convolutional layer and a zero-initialized 1D temporal convolutional layer. This produces a compact representation $\tilde{\mathbf{V}} \in \mathbb{R}^{(\frac{f}{2} \times \frac{h}{2} \times \frac{w}{2}) \times C}$. The output hidden state from the previous DiT block, denoted as $\mathbf{h} \in \mathbb{R}^{(\frac{f}{2} \times \frac{h}{2} \times \frac{w}{2}) \times C}$, is then combined with the trajectory latents ($\mathbf{V}$ and its group normalized output) before being forwarded to remaining DiT blocks: $\mathbf{h} = \mathbf{h} + \text{norm}(\tilde{\mathbf{V}}) + \tilde{\mathbf{V}}, \tilde{\mathbf{V}} = \text{Conv1D}(\text{Conv2D}(\text{patchify}(\mathbf{V})))$

**Loss Function** To learn the desired motion patterns, we optimize the parameters $\boldsymbol{\theta}$, including both the DiT blocks and the motion injector, as follows:

$$\mathcal{L}(\boldsymbol{\theta}) = \mathbb{E}_{\mathbf{x}, \mathbf{c}, \boldsymbol{\epsilon} \sim \mathcal{N}(\mathbf{0}, \sigma_t^2 \mathbf{I}), \mathbf{I}, \mathbf{M}_d, \mathbf{M}_s, \mathcal{C}, t} \left[ \| \boldsymbol{\epsilon} - \hat{\boldsymbol{\epsilon}}_{\boldsymbol{\theta}_1}(\mathbf{x}_t, \mathbf{c}, \mathbf{M}_d, \mathbf{M}_s, \mathcal{C}, t) \|_2^2 \right] \tag{6}$$

---

[2]We decompose the causal latent video $\mathbf{x}$ as three temporally-partitioned segments: $\mathbf{x}_1$, $\mathbf{x}_2$, and $\mathbf{x}_3$, corresponding to the pre-interaction, interaction, and post-interaction phases, respectively.

## 4 EXPERIMENTS

### 4.1 IMPLEMENTATION DETAILS

We implement our conditional video diffusion model based on the pre-trained CogVideoX-5B architecture (Yang et al., 2024b). We conduct experiments on the Bridge dataset (Walke et al., 2023) (Please refer to Sec. A.1), we adopt a resolution of $480 \times 640$ and a video length of 37 frames during both training and inference. The model is trained using AdamW (Loshchilov & Hutter, 2017) on 8 NVIDIA A800 GPUs, with a learning rate of $2 \times 10^{-5}$ for the DiT blocks and $1 \times 10^{-4}$ for the motion injector, and a total batch size of 16. Training is conducted for 30,000 steps. At inference, we employ 50 DDIM steps and set the CFG scale to 6.0.

### 4.2 BASELINES

We compare RoboMaster with existing state-of-the-art trajectory-controlled baselines: Tora (Zhang et al., 2025), MotionCtrl (Wang et al., 2024d), DragAnything (Wu et al., 2024) and IRASim (Zhu et al., 2024). For fair comparison, all baselines are retrained on the same dataset based on CogVideoX-5B with their respective optimal training configurations. We further compare with a SOTA 4D-grounded method, TesserAct (Zhen et al., 2025).

### 4.3 EVALUATION METRICS

We perform evaluation[3] on 214 test samples in Bridge, covering diverse manipulation skills[4], based on: 1) *Trajectory Accuracy*: We report the Trajectory Error (**TrajError**), which computes the average L1 distance between the input and generated trajectories of both the robot arm and the manipulated object. 2) *Video Quality*: We adopt standard metrics, including Frechét Video Distance (**FVD**) (Unterthiner et al., 2018), **PSNR** (Hore & Ziou, 2010), and **SSIM** (Wang et al., 2004).

### 4.4 QUANTITATIVE&QUALITATIVE COMPARISON

Table 2: **Quantative Comparison.** Note that all the baselines are retrained on our curated dataset.

| Method | Video Quality | | | Trajectory Accuracy | | User Study |
|---|---|---|---|---|---|---|
| | FVD ↓ | PSNR ↑ | SSIM ↑ | TrajError$_{robot}$ ↓ | TrajError$_{obj}$ ↓ | Preference ↑ (%) |
| TesserAct (Zhen et al., 2025) | 261.84 | 18.99 | 0.778 | 37.34 | 54.64 | 8.01 |
| IRASim (Zhu et al., 2024) | 159.04 | 20.88 | 0.782 | 19.25 | 34.39 | 6.45 |
| MotionCtrl (Wang et al., 2024d) | 170.79 | 19.89 | 0.761 | 21.17 | 28.52 | 9.68 |
| DragAnything (Wu et al., 2024) | 158.42 | 21.13 | 0.792 | 18.97 | 27.41 | 12.90 |
| Tora (Zhang et al., 2025) | 152.28 | 21.24 | 0.788 | 18.14 | 26.43 | 17.74 |
| **RoboMaster (Ours)** | **147.31** | **21.55** | **0.803** | **16.47** | **24.16** | **45.16** |

As shown in Fig. 5 and Tab. 2, RoboMaster consistently outperforms prior state-of-the-art methods in quantitative metrics of visual quality and trajectory accuracy, as well as in qualitative visual performance. Our strengths lie in two aspects: 1) *Interaction-aware Trajectory Design*: We explicitly decompose interaction phases and integrate object features into a unified trajectory. In contrast, baseline methods struggle with feature entanglement in regions where the robotic arm and object interact. IRASim only controls the robot trajectory, resulting in coarse object control and increased trajectory error ($24.16 \rightarrow 34.39$). 2) *Object Representation*: We use mask-based representations rather than shape-ambiguous point representations (as in Tora and MotionCtrl), leading to improved object identity consistency across frames. See the white-box region in Fig. 5 for a comparison of object identity preservation. RoboMaster further exhibits enhanced robustness on in-the-wild image collections, outperforming baseline methods as shown in Fig. 6.

---

[3]Generate a video based on an initial frame, a prompt, robot and object trajectories, and an optional object mask (for Tora and Ours), and then compare it with GT video.

[4]Skills: move, pick, open, close, upright, topple, pour, wipe, and fold

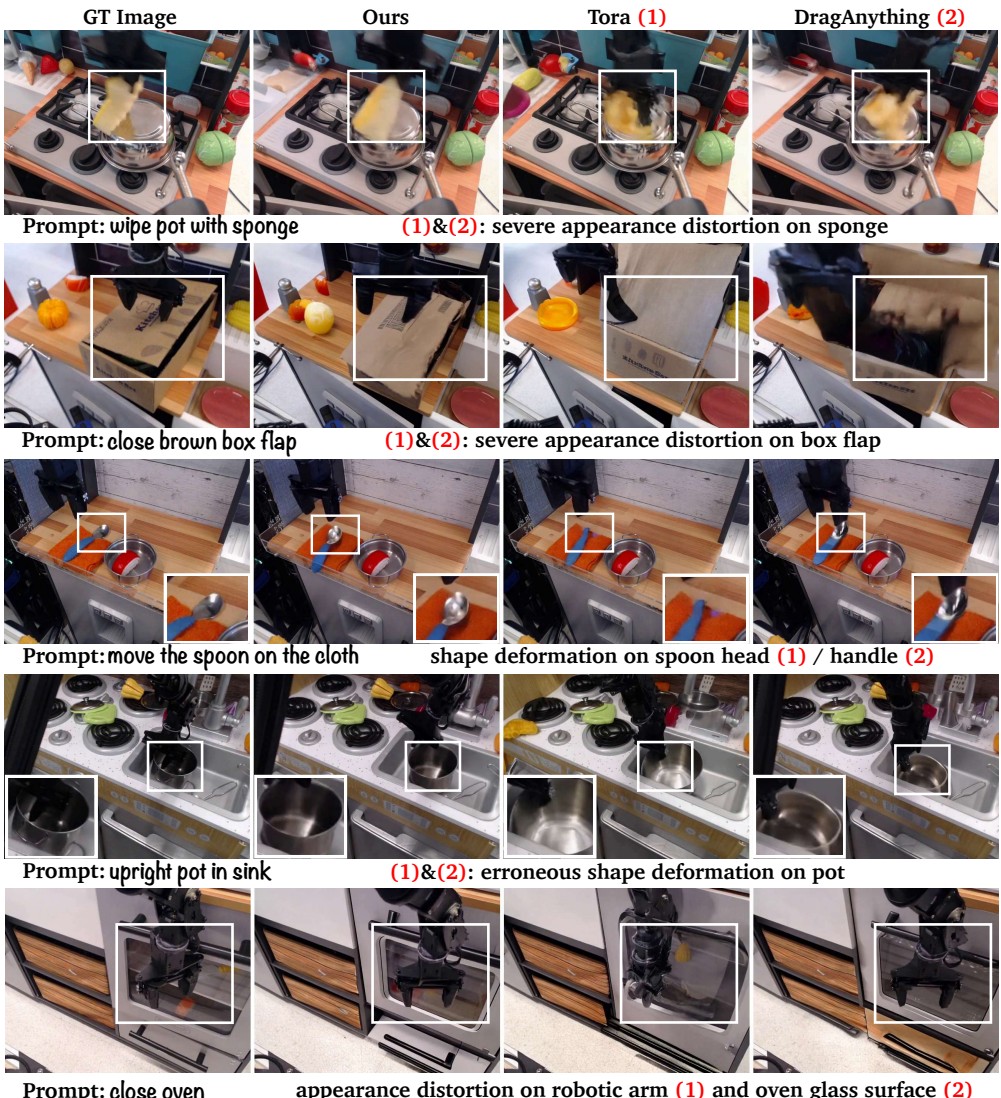

Figure 5: **Qualitative Comparison.** RoboMaster demonstrates superior performance across a range of manipulation skills (e.g., move, pick, close, upright, close), exhibiting improved visual consistency of the manipulated subject compared to prior baselines.

## 4.5 EMBODIED ACTION PLANNING FOR ROBOTIC POLICY

For robotic planning simulation, we adopt five challenging tasks from RLBench (James et al., 2020) ("pick up cup", "put knife", "put plate", "open microwave", and "close box") and four tasks from the SIMPLER benchmark (Li et al., 2024) ("pick coke can", "close drawer", "move near", and "pick object"). For baselines, We adopt Tora (Zhang et al., 2025), TesserAct (Zhen et al., 2025) and OpenVLA (Kim et al., 2024) and also finetune them on the same curated dataset for fair comparison.

The general pipeline for applying RoboMaster in robot learning is as follows: 1) Generate demonstration videos given the first frame and the robotic task prompt using RoboMaster. 2) Extract executable action labels from the generated demonstrations. 3) Simulate robotic planning and evaluate the task success rate.

For the first stage, we collect a small set of 1,300 video–trajectory pairs from both benchmarks, ensuring that none of the corresponding tasks appear in the test set. Here we fix the camera location in each task. We then post-train RoboMaster on this limited dataset to adapt the model to the benchmark robot morphologies (Franka in RLBench and Google Robot in SIMPLER). The model architecture is kept unchanged and training is performed on 8 NVIDIA A800 GPUs for $\sim$ 6 hours. After this

adaptation, RoboMaster is able to generate demonstrations conditioned on the benchmark robot morphologies.

or the second stage, we collect 300 video-action samples for each task in testset to train the inverse dynamic model. Following AVDC (Ko et al., 2023), the inverse model is trained to regress executable action labels from synthesized videos. To further validate its effectiveness, we additionally post-train the Cosmos-Predict2.5-2B/robot/action-cond model from Cosmos2.5 (Ali et al., 2025) on the curated training data. The model takes as input the first frame and a sequence of 7-DoF actions ( $\Delta x, \Delta y, \Delta z, \Delta\theta_r, \Delta\theta_p, \Delta\theta_y$, GripperWidth), and outputs the predicted video. We perform fully fine-tuning, resizing videos to $320 \times 256$ with 37 frames, using a learning rate of 2e-5, training for 2,500 iterations with a batch size of 16. Then we feed actions predicted by our inverse dynamic model into this model to generate action-conditioned videos. Quantitative evaluation against GT videos, as shown in Tab. 3, demonstrates that actions predicted by our inverse dynamics model produce comparable video quality, further validating its high performance.

Table 3: **Evaluation of Action-conditioned Videos**

| Action Type | PSNR ↑ | SSIM ↑ | Latent L2 ↓ | FVD ↓ |
|---|---|---|---|---|
| Training Set | 25.48 | 0.87 | 0.31 | 132 |
| Predicted | 25.12 | 0.84 | 0.34 | 127 |

For the final evaluation stage, we generate 100 videos per task using the RoboMaster adapted in Stage 1, conditioned on the task prompt and the initial frame. We then apply the inverse dynamics model to infer executable action labels from these generated videos and deploy the resulting action sequences on simulated robots. The success rates over 100 trials are summarized in Tab. 4. RoboMaster consistently improves embodied action planning over existing baselines. The clear margin by both Tora and RoboMaster over OpenVLA demonstrates that high-quality video generations serve as effective demonstrations for downstream robotic policy extraction. Moreover, RoboMaster outperforms Tora on 8 out of the 10 evaluated tasks, suggesting that RoboMaster produces more reliable interaction videos, enabling the inverse dynamics model to obtain more accurate action labels for execution. This further validates our core motivation: more accurate modeling of robot–object interactions leads to higher-quality execution demonstrations for robot learning.

Table 4: **Action Planning Comparision on RLBench and SIMPLER.** We report the success rate averaged over 100 episodes for each task. Best is bolded and second best is underlined.

| Method | RLBench | | | | | SIMPLER | | | |
|---|---|---|---|---|---|---|---|---|---|
| | pick up cup | put knife | put plate | open microwave | close box | pick coke can | close drawer | move near | pick object |
| OpenVLA (Kim et al., 2024) | 0.55 | 0.46 | 0.56 | 0.35 | 0.45 | 0.59 | 0.41 | 0.53 | 0.59 |
| TesserAct (Zhen et al., 2025) | 0.76 | 0.79 | 0.82 | 0.43 | 0.67 | 0.85 | 0.56 | 0.62 | 0.79 |
| Tora (Zhang et al., 2025) | 0.79 | **0.82** | 0.81 | **0.61** | 0.72 | 0.89 | 0.61 | 0.61 | 0.74 |
| **RoboMaster** | **0.83** | 0.76 | **0.85** | 0.54 | **0.79** | **0.91** | **0.63** | **0.67** | **0.81** |

## 4.6 ABLATION STUDY

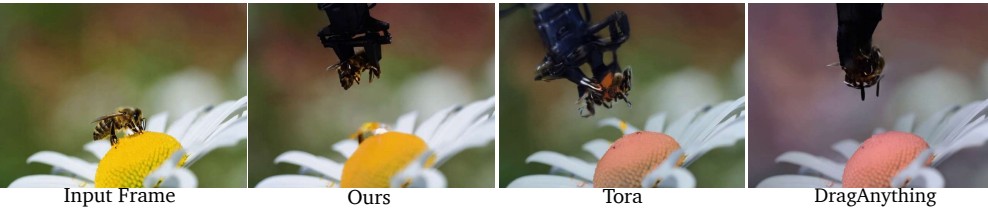

| Input Frame | Ours | Tora | DragAnything |

Figure 6: **Generalizable Comparison with Input Prompt**: 'Pick up the bee.'

We perform ablation on the full evaluation benchmark to validate model component effectiveness.

**Subject Representation.** Removing the causal embedding for latent control (w/o Causal Embedding in Tab. 5) leads to a decline in both visual quality and trajectory accuracy, as evidenced by the misplacement of the can in Fig. 7, highlighting the necessity of conditioning causal visual latents on

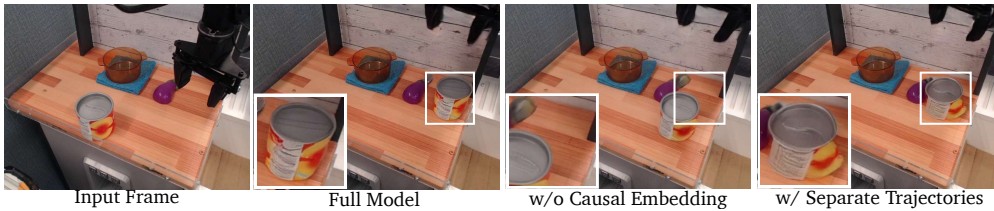

Input Frame      Full Model      w/o Causal Embedding      w/ Separate Trajectories

Figure 7: **Ablation on a Generalizable Sample**: 'Move the can to the right place of the eggplant.'

Table 5: **Ablation Study on Bridge Full Benchmark.**

| Method | FVD ↓ | PSNR ↑ | SSIM ↑ | TrajError$_{robot}$ ↓ | TrajError$_{obj}$ ↓ |
|---|---|---|---|---|---|
| w/o Causal Embedding | 151.62 | 21.30 | 0.797 | 18.32 | 27.15 |
| w/ Points Representation | 157.49 | 20.87 | 0.779 | 19.71 | 31.41 |
| w/ Separate Trajectories | 152.01 | 21.08 | 0.792 | 17.24 | 25.84 |
| w/ Cross Attention | 163.56 | 19.38 | 0.761 | 21.52 | 29.16 |
| **Full Model** | **147.31** | **21.55** | **0.803** | **16.47** | **24.16** |

Table 6: **Ablation on Mask Sparsity**

| Sparsity (%) | PSNR (%) |
|---|---|
| 90 | 99.81 |
| 80 | 98.12 |
| 70 | 98.02 |
| 60 | 97.89 |

causal control signals. Moreover, replacing the mask-based representation with a point-based one (w/ Point Representation), as in Tora, significantly increases the subject trajectory error ($24.16 \rightarrow 31.41$), indicating that mask provides a more effective object representation. The mask-based approach also exhibits greater robustness to input sparsity, as shown in Tab. 6, where PSNR is reported relative to the full-mask baseline—an important property for real-world user input that is often incomplete.

**Trajectory Injection.** Replacing the collaborative trajectory with separate ones (w/ Separate Trajectories) introduces feature fusion issues in overlapping regions, leading to reduced visual quality (see can distortion in Fig. 7) and lower trajectory accuracy. This supports the effectiveness of our decomposed trajectory design. Moreover, our model remains simple and effective, as alternative designs such as cross-attention-based trajectory injection (w/ Cross Attention) result in degradation. When randomly deviating a partial subset ($\sim 15\%$) of sampled points from the original trajectory, the generated video remains robust under such disturbances as shown in Tab. 7.

Table 7: **Ablation on Trajectory Perturbation**

| Deviration (%) | PSNR (%) |
|---|---|
| 5 | 99.17 |
| 10 | 98.25 |
| 15 | 97.68 |
| 20 | 97.15 |

**Imperfect Prompts.** We further analyze the sensitivity of our model to inaccurate prompts. Specifically, we randomly replace the subject prompt with prompt of similar or entirely different semantics (e.g., "a yellow sponge"→"a yellow block" or "cotton ball"; "a spoon"→"a branch") using GPT-4. Experiments conducted on the full Bridge benchmark produce the results shown in Tab. 8, where PSNR is reported relative to the accurate-prompt baseline. Even with 40% of the prompts replaced by imperfect descriptions, RoboMaster retains over 96% of its full-prompt performance, demonstrating strong robustness to prompt inaccuracies.

Table 8: **Ablation on Imperfect Prompt Input**

| Erroneous Portion (%) | PSNR (%) |
|---|---|
| 10 | 98.42 |
| 20 | 97.53 |
| 30 | 97.13 |
| 40 | 96.54 |

## 5 CONCLUSION

In this work, we present RoboMaster, a trajectory-controlled video generation framework with a collaborative interaction design tailored for robotic manipulation. By decomposing interactions into sub-interaction phases, our method achieves superior visual quality and trajectory accuracy over prior approaches. Coupled with shape- and appearance-aware object encoding, RoboMaster enables more intuitive user annotation and enhances overall interactivity.

**Limitations**&**Future Work**: (1) RoboMaster may produce incomplete or distorted objects during manipulation when applied to out-of-domain inputs. This could be mitigated by training on more diverse object categories with richer semantic and geometric variations. (2) The current framework operates purely in 2D pixel space; integrating depth cues (Fu et al., 2024; Chen et al., 2025; Hu et al., 2024) may enable more accurate 3D control. (3) Generalization to varied robotic embodiments remains a challenge and requires expanding training data to encompass broader robot configurations.

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
