## APPENDIX

## A    EXPERIMENTAL DETAILS

### A.1    DATASET CURATION

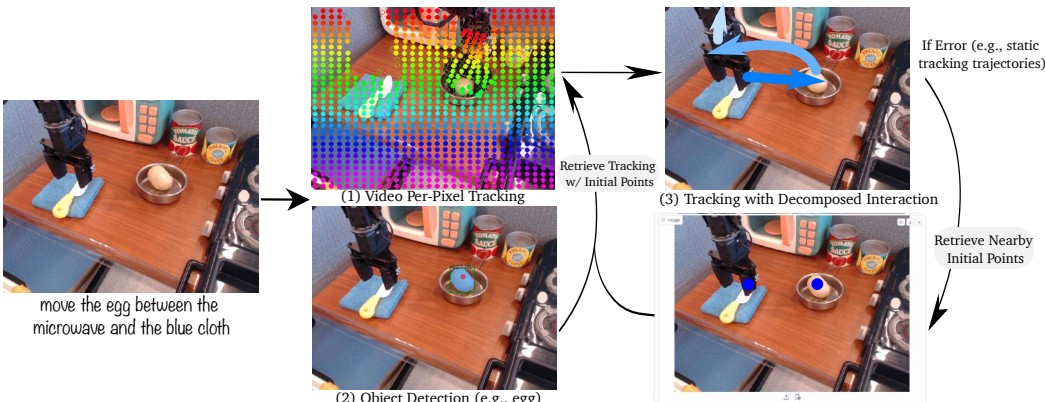

Figure S8: **Dataset Construction Pipeline.** It involves automatic annotation with human-in-the-loop processes to generate high-quality data samples.

Given a raw video $\mathbf{X}$ paired with a prompt $\mathbf{c}$, we generate the annotations following the stream below:

(1) *Video Per-Pixel Tracking*: We employ CoTracker3 (Karaev et al., 2024) to compute spatio-temporal trajectories of point sets, which are initialized on a dense grid interval (30) in the first frame.

(2) *Object Detection*: We parse the prompt to extract the noun corresponding to the submissive object $\mathbf{o}_s$, typically the first noun following the action verb. The dominant object $\mathbf{o}_d$ is either parsed or pre-defined (e.g., 'black robotic gripper' in Bridge (Walke et al., 2023)). We apply Grounded SAM (Ren et al., 2024) to obtain segmentation masks of interacting entities in the first frame, compute their centers of gravity, and associate them with the nearest tracking point in the first frame and its tracking trajectory in step (1).

(3) *Decoupling Interaction*: To identify the transition frames marking the start and end of the interaction phase ($F_1$, $F_2$), we analyze the motion dynamics of the submissive object throughout the video. Transitions are determined by applying a motion threshold $\tau$ to detect the timestamps where the object initiates and terminates its activity.

As shown in Fig. S8, we apply the automatic annotation pipeline to each video in the training set and filter out invalid samples resulting from failures in object detection or trajectory tracking. On the Bridge dataset (Walke et al., 2023), this process yields approximately 21k annotated video samples.

### A.2    USER ANNOTATIONS ON IN-THE-WILD IMAGES

To facilitate user-friendly annotation on in-the-wild image samples, we develop a Gradio demo, as shown in Fig. S9. This interactive interface requires the user to provide the following inputs, which are prepared for the model:

(1) *Text Prompt*: Describing the interaction type (e.g., pick, move) and target manipulated object.

(2) *Object Mask*: The user employs the brush tool to define the region of the manipulated object. Note that the user only needs to provide the object mask, while the robotic arm mask is pre-defined and can be used as an off-the-shelf component.

(3) *Time Period of Interaction*: The user specifies the start and end timestamps for the interaction. If the interaction does not include a post-interaction phase (e.g., pick, open and close), the end timestamp is set to the maximum video length.

Table R9: **Summary of Annotated Transition Cases.**

| Case | Transition Point (1) | Transition Point (2) | Example |
|:---:|:---:|:---:|:---:|
| 1 | ✓ | ✓ | pick up the lemon and place it on the table |
| 2 | ✓ | - | pick up the lobster |
| 3 | - | ✓ | rotate the sponge and place it |
| 4 | - | - | rotate the sponge |

(4) *Collaborative Trajectory*: Annotate key points on the input image for each decomposed interaction phase, and the completed trajectory is generated through interpolation. Take this picture as example: In the upper-right panel, user can annotate only four blue points as key points to sample a full trajectory in interaction phase via interval interpolation. In the upper-left panels, the red point indicates the transition from pre-interaction to interaction, while the green point marks the transition from interaction to post-interaction. To refine the trajectory definition, we visualize the intermediate images and composite video after each input is completed. In practice, users do not need to mark all transition points. Trajectories can be categorized into four cases, as summarized in Tab. R9, where Transition Point (1) indicates pre-interaction $\rightarrow$ interaction, and Transition Point (2) indicates interaction $\rightarrow$ post-interaction. These four cases cover the majority of practical generalization scenarios.

## A.3 Network Architecture

As shown in Tab. R10, the architecture of RoboMaster incorporates the collaborative trajectory latent $\mathbf{V}$ into the base model to facilitate the visual generation of the robotic manipulation video $\mathbf{X}$.

Table R10: **Network Architecture.** $N$, $C$, and ks denote the block number in the base video model, the latent feature size, and the kernel size in each 2D/1D convolutional layer, respectively.

| Input | Layer | Output | Output Dimension |
|:---:|:---:|:---:|:---:|
| Image $\mathbf{I}$ | - | - | $H \times W \times 3$ |
| Image $\mathbf{I}$ | VAE ($\mathcal{E}(\cdot)$) | $\mathbf{z}$ | $h \times w \times c$ |
| $\mathbf{z}$ + init. noise | Patchify | $\mathbf{h}$ | $(f/2 \times h/2 \times w/2) \times C$ |
| Collab. Traj. Latent $\mathbf{V}$ | - | - | $f \times h \times w \times c$ |
| $\mathbf{V}$ | $\left( \begin{array}{l} \text{Conv2D (ks=3, } C_{\text{in}} = 8c, C_{\text{out}} = C/4, \text{ padding } = 1), \\ \text{Conv1D (ks=3, } C_{\text{in}} = C/4, C_{\text{out}} = C, \text{ padding } = 1), \\ \text{FloatGroupNorm } (n_{\text{groups}} = 32, C_{\text{out}} = C), \end{array} \right) \times N$ | $\tilde{\mathbf{V}}$ | $(f/2 \times h/2 \times w/2) \times C$ |
| $\mathbf{h} + \tilde{\mathbf{V}}$ | $\left( \begin{array}{l} \text{LayerNorm + 3D Attention }, \\ \text{LayerNorm + Feed-Forward}, \end{array} \right) \times N$ | $\mathbf{h}$ | $(f/2 \times h/2 \times w/2) \times C$ |
| $\mathbf{h}^{t=0}$ | Unpatchify + VAE ($\mathcal{D}(\cdot)$) | $\mathbf{X}$ | $F \times H \times W \times 3$ |

## B More Analysis on Limitations

**Control Granularity in 3D Space.** Incorporating 3D cues may further improve the success rates, but it also encounters challenges: 1) *3D feature entanglement*: Even when 3D cues (e.g., depth) are integrated, prior works (Tora, DragAnything, and MotionCtrl) still face feature entanglement issue during interaction in 3D space. Handling 3D occlusions and spatial configurations presents an additional challenge for accurate 3D feature modeling. 2) *User annotation burden*: From the user's perspective, incorporating additional z-dimension annotations may introduce nontrivial labeling burden, especially when depth estimation is unreliable. However, extending our framework towards fully 3D-aware interaction remains a promising direction for more precise control.

**Usage of Automatic Segmentation Models.** Integrating automatic grounding or segmentation methods can accelerate inference when scaling up data generation. However, two practical challenges remain: 1) *Multiple-object Scenarios*: When the input image contains complex scenes or multiple instances of the same category (e.g., several nearly identical avocados, dumplings, lobsters, mushroom, red peppers, or strawberries, as shown under "Robotic Manipulation on Diverse Out-of-Domain Objects" on our website), current models still struggle to reliably identify the target object, even with

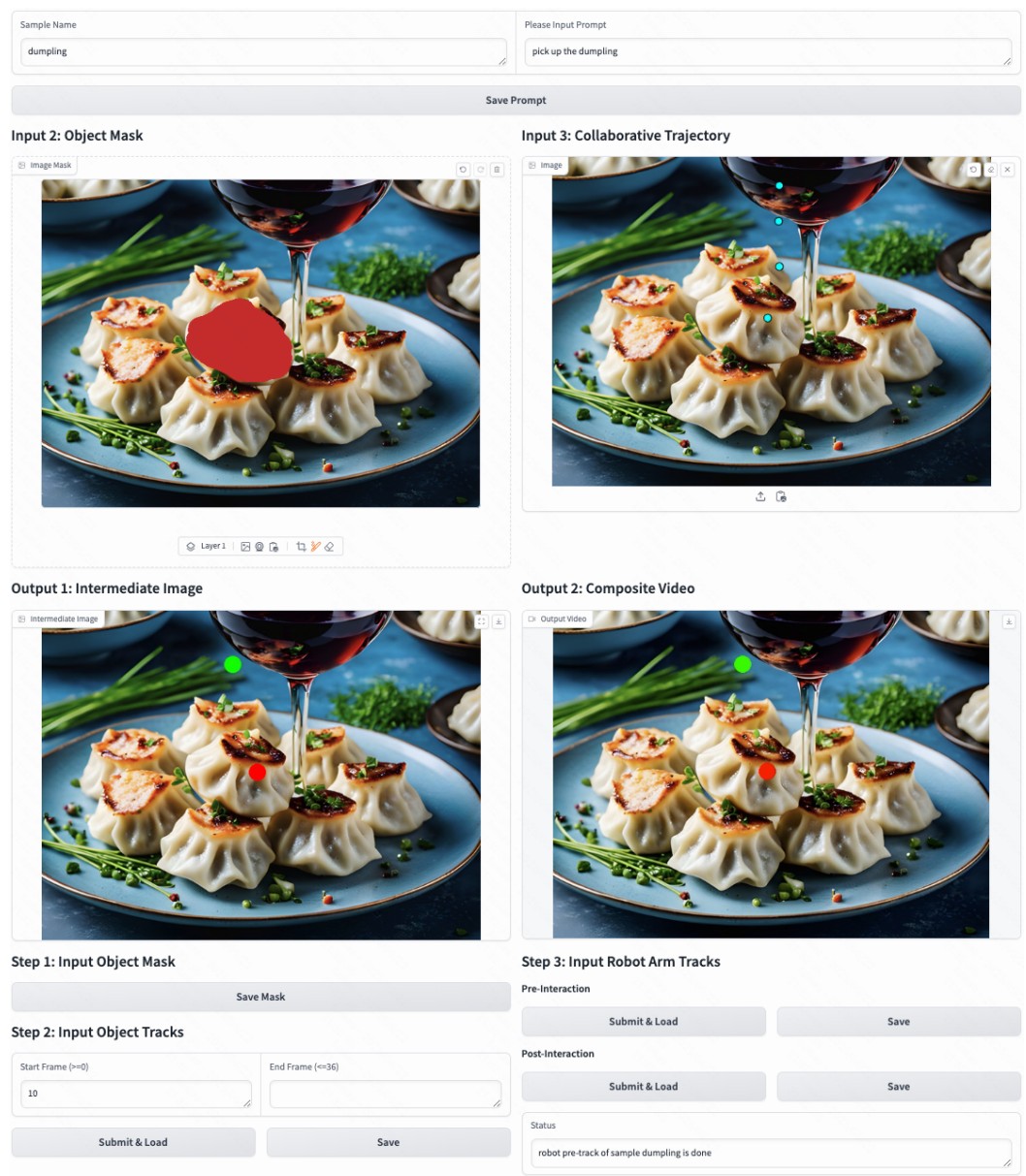

Figure S9: **Gradio Demo for User Annotation.** The user is required to provide a prompt, an object mask, and a collaborative trajectory consisting of three phases: pre-interaction, interaction, and post-interaction, in sequence. This setup allows for flexible edits at any stage, enabling iterative refinement of the annotation.

detailed prompt descriptions, as Grounded SAM achieves only a 0.14 success rate on the training set and 0.21 on the in-the-wild set as shown in Tab. R11. 2) *Single-object Scenarios*: Even in simpler single-object scenarios, these models are still not fully reliable. During the annotation of our training data as described in Sec. A.1, we initially employed Grounded SAM (Ren et al., 2024) to generate object masks. However, the first annotation pass yielded only 17,000 reliable masks out of 25,000 videos. And we further manually re-annotate for approximately 4,000 videos. A detailed success rate of applying Grounded-SAM based annotation is shown in Tab. R11. These observations indicate that manual annotation prior may remain necessary for some complex scenes, whereas automatic methods can still be effective in simpler cases.

**Out-of-Domain Generalization Ability.** The generalization capability of prior video-generation methods (Zhu et al., 2024; Ko et al., 2023; Yang et al., 2023; Du et al., 2023) for robotic manipulation

Table R11: **Success Rate of Grounded-SAM based Annotation.**

| Dataset Type | Data Number | Single-Object | Multiple-Objects |
|---|---|---|---|
| Training Set | 25,421 | 0.67 | 0.14 |
| In-the-wild | 214 | 0.75 | 0.21 |

is largely constrained to scenarios closely aligned with their training distributions. In contrast, RoboMaster exhibits substantially stronger out-of-domain generalization across diverse unseen object categories and backgrounds (e.g., bee, cartoon bottle, dumpling, lobster). Representative videos are provided under "Robotic Manipulation on Diverse Out-of-Domain Objects" on our website. Besides, our model benefits not only from the diversity of training data during post-training, but also from the strong priors encoded in the pretrained video backbone. We believe that a base model (Ali et al., 2025; Agarwal et al., 2025; Wang et al., 2025) endowed with richer physical and interaction priors would further enhance generalization under the same amount of task-specific training data.

**Generalization to Novel Robot Morphologies.** It still remains a challenging unresolved problem. Even recent work such as TesserAct (Zhen et al., 2025), which supports multiple robot morphologies including Google Robot, WidowX, and Franka Panda, relies on morphologies that are all present in the training dataset. It still cannot generalize to unseen robots, such as xArm or MobileALOHA. Achieving generalization to novel morphologies may require backbone pretraining or post-training on more diverse datasets and robot configurations. Another potential approach is to leverage a single or few demonstration videos at test time and a lightweight adaptation network (e.g., LoRA (Hu et al., 2022)) to acquire a new robot morphology during inference.

## C ADDITIONAL RELATED WORK

**Trajectory-Controlled Video Generation.** Recent advances in trajectory-conditioned video generation primarily fall into two directions: *Camera Movement*: MotionCtrl (Wang et al., 2024d), CameraCtrl (He et al., 2024), and 4DiM (Watson et al., 2024) have successfully implemented camera-controlled text-/image-to-video generation using 6-DoF camera trajectories. NVS-Solver (You et al., 2024) enhances generalizability by employing training-free depth-warping during the denoising process. ReconX (Liu et al., 2024a) and ViewCrafter (Yu et al., 2024b) improve 3D consistency by projecting point clouds into a 3D cached space for guidance. CVD (Kuang et al., 2024) and SynCamMaster (Bai et al., 2024) expand camera control to multi-shot generation. VD3D (Bahmani et al., 2024b) and AC3D (Bahmani et al., 2024a) integrate camera control into DiT-based video generation models. Additionally, recent studies (Bai et al., 2025; Ren et al., 2025; YU et al., 2025) explore re-capturing a source video using a specified camera trajectory. In contrast to these approaches, RoboMaster emphasizes collaborative object trajectory control rather than focusing on camera trajectory. 2) *Object Movement*: refer to main paper.

**Video Generation with Injected Control.** *(1) Training-free Approaches*: These methods directly manipulate attention patterns or latent representations at the inference time, though constrained by limited generalizability and demanding empirical tuning. Direct-a-Video (Yang et al., 2024a) modulates spatial cross-attention maps under the guidance of a bounding box. FreeTraj (Qiu et al., 2024) implements spectral-domain trajectory embedding with attention reweighting. DiTCtrl (Cai et al., 2024) convert self-attention into the proposed masked-guided KV-sharing strategy to generate multi-prompt video. MOFT (Xiao et al., 2024) employs motion-channel disentanglement and sampling process modification through reference-based priors. *(2) Learning-based Approaches*: Previous techniques typically employ auxiliary encoders to map control signals into latent representations, utilizing learnable components (e.g., convolutional/linear layers, attention modules, LoRA adapters) or leveraging frozen pre-trained feature extractors. These encoded features are subsequently fused with the base model through feature fusion techniques such as concatenation, additive merging, or cross-attention injection. VideoComposer (Wang et al., 2023) employs a unified STC-encoder and CLIP model to condition the base T2V model with multi-modal input conditions. MotionCtrl (Wang et al., 2024d) introduces object motion control via an additional motion encoder. SparseCtrl (Guo et al., 2024) learns an add-on encoder to integrate various control signals into the base model. Tora (Zhang et al., 2025) employs a trajectory encoder and plug-and-play motion fuser to merge

2D trajectories with the base video model. MotionDirector (Zhao et al., 2024) leverages spatial and temporal LoRA layers to learn desired motion patterns from reference videos. Motion Prompting (Geng et al., 2024b) excels in various controllable generation tasks via training a ControlNet-style adapter with general motion conditions. Meanwhile, a line of works (Hu, 2024; Tan et al., 2024; Gan et al., 2025) designs sophisticated control mechanisms for human animation.

# D   ADDITIONAL QUANTATIVE&QUALITATIVE RESULTS

## D.1   VBENCH COMPARISION

We further evaluate video quality using the widely adopted VBench (Huang et al., 2024). As shown in Tab. R12, RoboMaster outperforms baselines across diverse evaluation metrics.

Table R12: **Quantative Comparison on VBench (Huang et al., 2024) Metrics.**

| Method | Aesthetic Quality ↑ | Imaging Quality ↑ | Temporal Flickering ↑ | Motion Smoothness ↑ | Subject Consistency ↑ | Background Consistency ↑ |
|---|---|---|---|---|---|---|
| IRASim (Zhu et al., 2024) | 50.12 | 67.11 | 98.04 | 98.79 | 93.11 | 94.89 |
| MotionCtrl (Wang et al., 2024d) | 48.78 | 66.78 | 98.21 | 97.58 | 92.19 | 95.15 |
| DragAnything (Wu et al., 2024) | 49.53 | 67.15 | 97.83 | 98.25 | 93.01 | 95.14 |
| Tora (Zhang et al., 2025) | **50.61** | 67.28 | 97.79 | 98.11 | 92.71 | 95.26 |
| **RoboMaster (Ours)** | 50.32 | **67.49** | **98.27** | **98.81** | **93.55** | **95.40** |

## D.2   ROBOTIC MANIPULATION ON DIVERSE OUT-OF-DOMAIN OBJECTS

As demonstrated in Fig. S10, RoboMaster is capable of generalizing to a wide range of in-the-wild objects, such as bee, bottle, and peach in the oil painting, as well as dumpling, lobster, pumpkin head, and teddy bear, despite being trained solely on the Bridge dataset.

## D.3   ROBOTIC MANIPULATION WITH DIVERSE SKILLS

As shown in Fig. S11, RoboMaster demonstrates the ability to perform a wide range of manipulation tasks on real-world image datasets, including pick, pick-and-place, move, open, close, topple, fold, upright, and wipe.

## D.4   LONG VIDEO GENERATION IN AUTO-REGRESSIVE MANNER

Robomaster facilitates the generation of extended videos in an auto-regressive manner. Specifically, given either the initial frame or the final frame of a previously generated video, it progressively generates a longer, coherent video by utilizing multiple ordered prompts, as illustrated in Fig. S12.

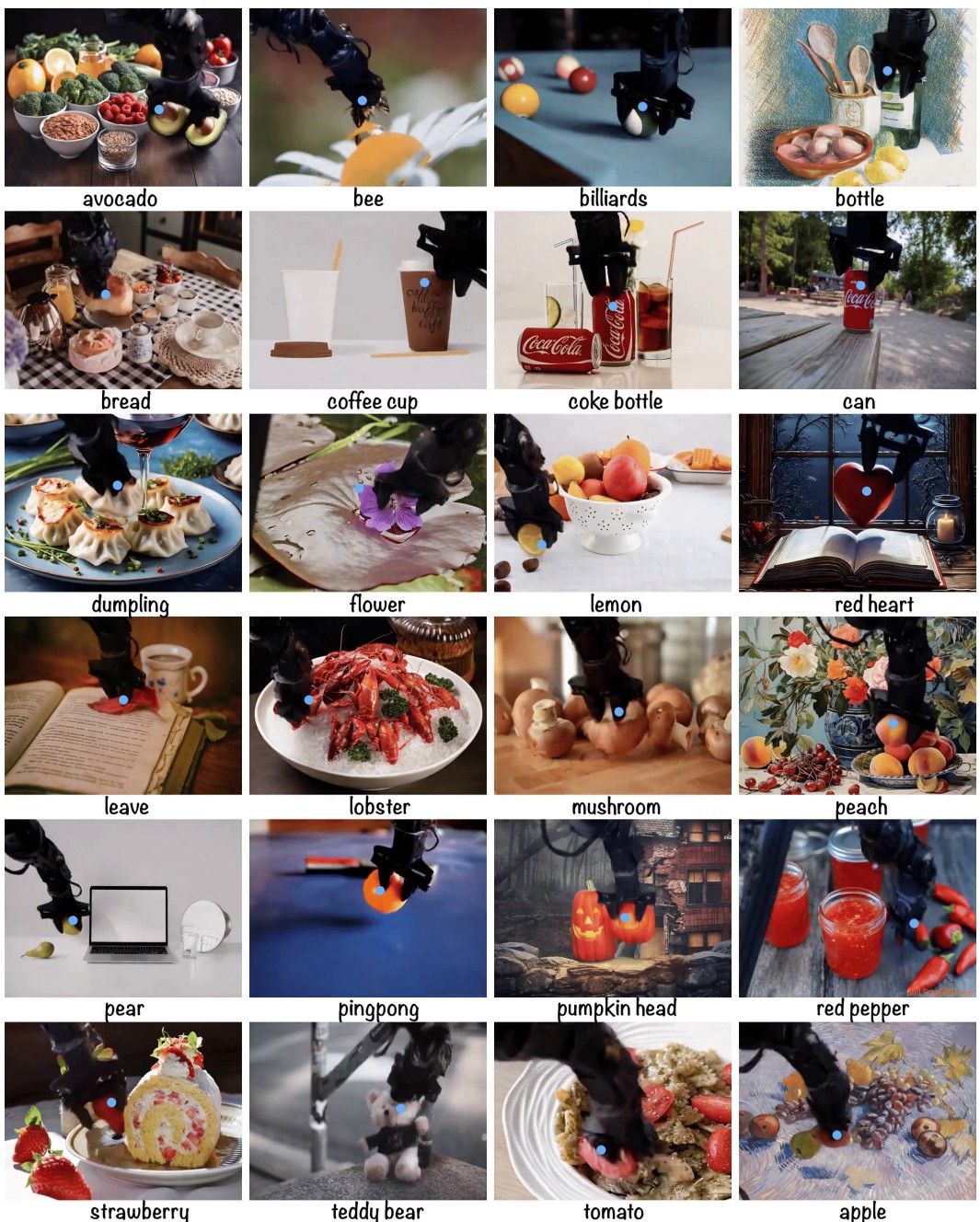

Figure S10: **'Pick up' on Diverse Out-of-domain (OOD) Objects.** The blue dot represents the current position of the manipulated object along the guided trajectory.

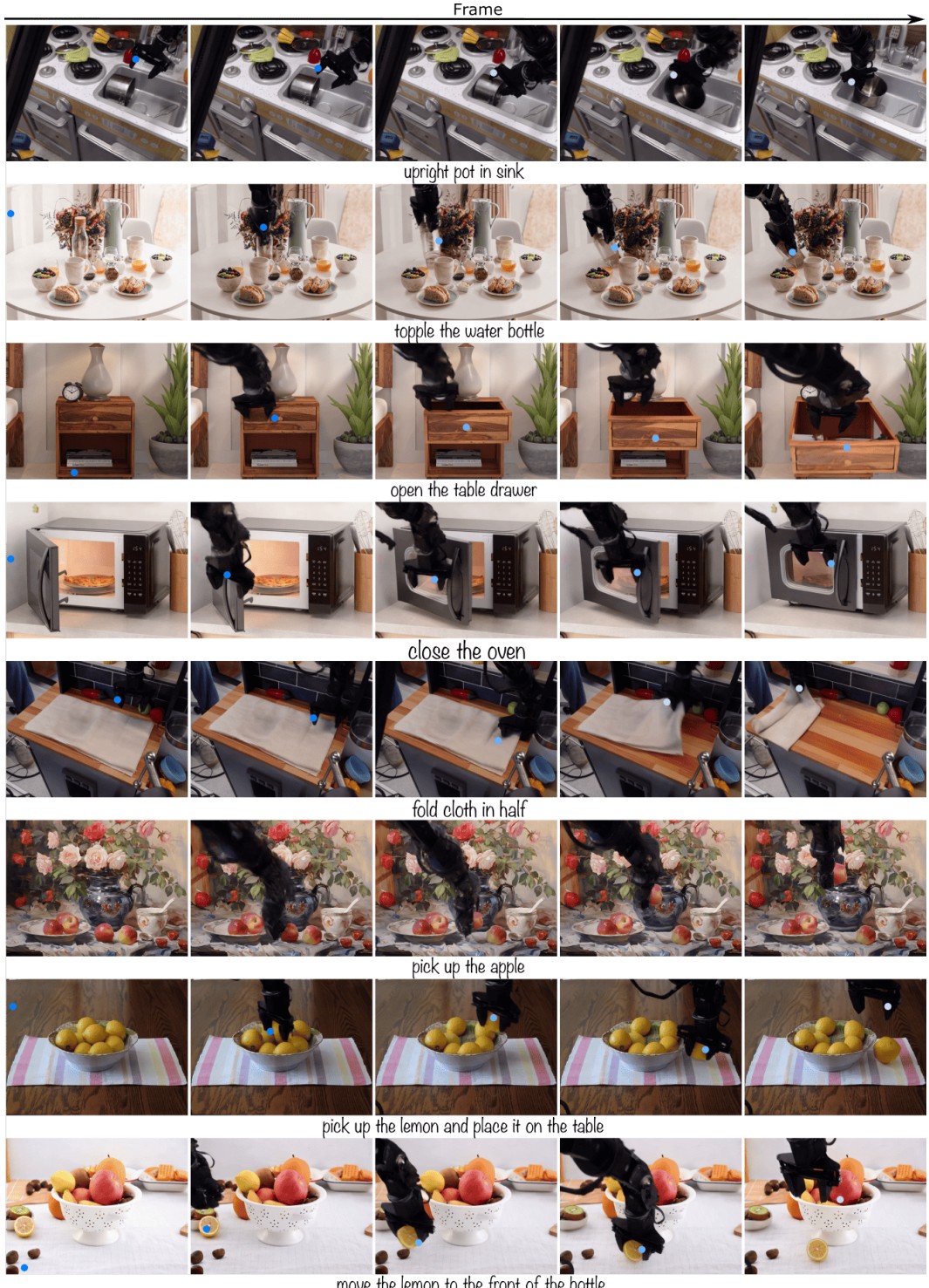

Figure S11: **Diverse Manipulation Skills on Bridge and In-the-wild Test Samples.**

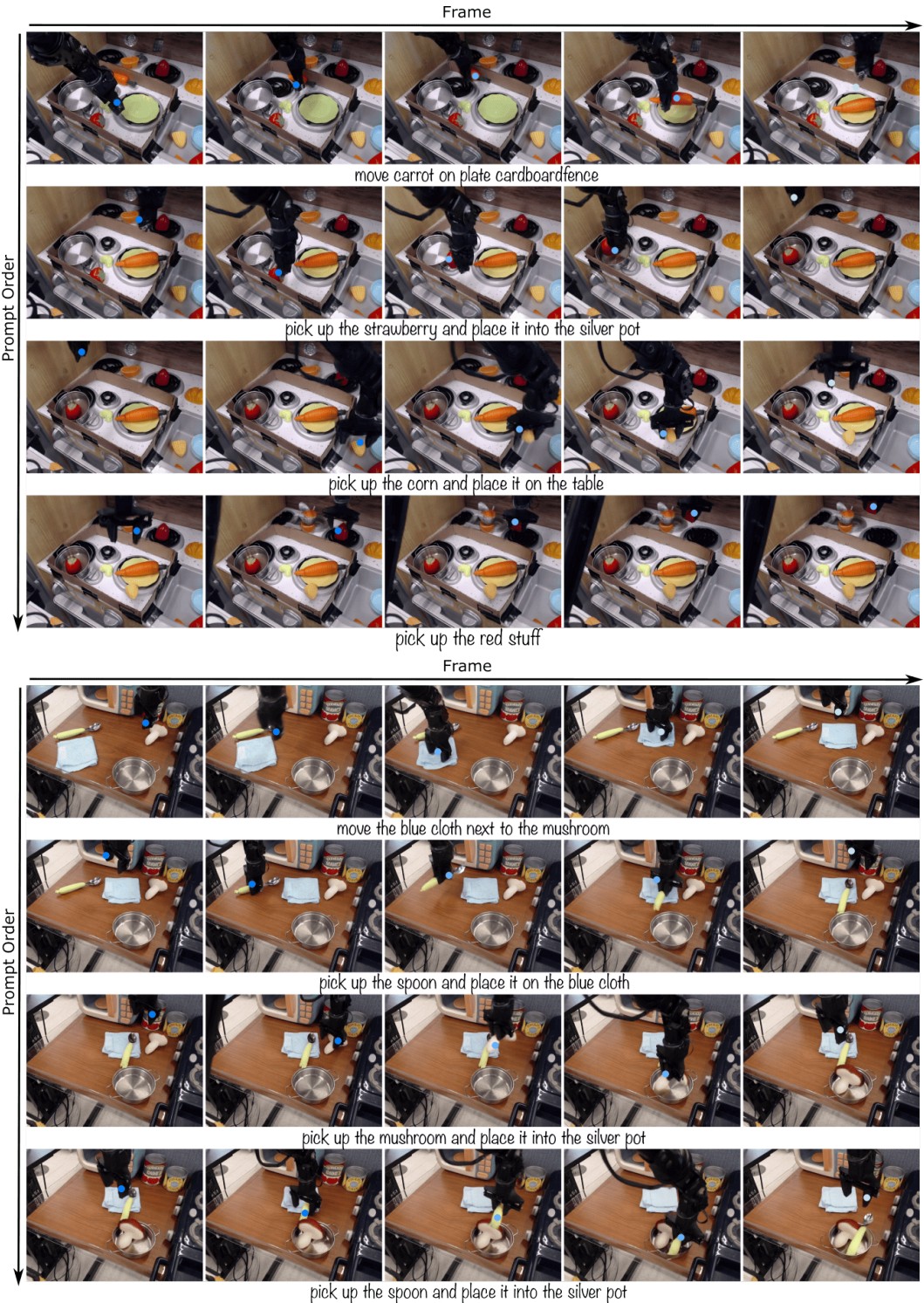

Figure S12: **Longer Video Generation with Multiple Input Prompts.**