# OpenReview forum: "Learning Video Generation for Robotic Manipulation with Collaborative Trajectory Control"
_ICLR.cc/2026/Conference — ICLR 2026 Poster_

### Official Review · Reviewer_1NRP · 2025-10-24

**Soundness:** 3
**Presentation:** 3
**Contribution:** 3
**Rating:** 6
**Confidence:** 3

**Summary:**

Summary :

RoboMaster is a trajectory‑conditioned video diffusion framework for robotic manipulation that jointly controls the robot arm and the object with a single collaborative trajectory. By decomposing each task into pre‑interaction, interaction, and post‑interaction phases and using mask‑based, appearance‑ and shape‑aware embeddings, it avoids the feature entanglement seen when controlling objects separately. It achieves state‑of‑the‑art visual fidelity and trajectory accuracy and improves downstream action planning.


Contributions:

Collaborative trajectory control for interaction: A unified trajectory that models pre-interaction → interaction → post-interaction phases, switching the dominant controller (arm → object → arm) to capture inter-object dynamics and avoid feature fusion during contact.

Appearance and shape aware subject embeddings: Mask-based tokens drawn from the initial frame’s VAE latents, expanded into shape-aware circular volumes along the trajectory to preserve object identity across frames.

Causal latent propagation: Frame-to-frame latent carryover (with overwrite at trajectory points) for smoother, temporally consistent motion during generation.

**Strengths:**

1. Novel collaborative trajectory design

Introduces a new way to model robot–object interactions using a single collaborative trajectory split into pre-interaction, interaction, and post-interaction phases.

Avoids the feature entanglement issues (e.g., missing or distorted objects) that plague prior methods like Tora and DragAnything.

2. High visual and physical realism

Produces smoother, more physically plausible manipulation videos with consistent object identities across frames.

Quantitatively achieves better FVD, PSNR, and SSIM, and lower trajectory errors on the Bridge benchmark.

3. Generalization and robustness

Handles diverse manipulation skills and in-the-wild scenarios.

Robust to imperfect user input—works with coarse or partial object masks and noisy trajectories.

**Weaknesses:**

1. Restricted to 2D pixel space

The system does not yet model depth or 3D geometry; this limits physical accuracy and makes 3D control (e.g., precise grasping) difficult.

2. Possible failure on out-of-domain inputs

Can produce incomplete or distorted objects when encountering unseen categories or backgrounds.

Still relies on training data diversity to generalize effectively.

3. Semantic dependency on user input

Relies on accurate prompts and roughly correct masks. Misleading text or poor masks may still degrade quality.

**Questions:**

Question 1: How does the model determine the precise transition points between these phases in practice, especially when the temporal boundaries of “interaction” are ambiguous or vary across tasks?


Question 2: How well would RoboMaster generalize to unseen robot morphologies or entirely different kinematic structures, and what adaptations (e.g., in trajectory representation or latent space) would be needed for that?

---

> ### Author Response · Authors · 2025-11-22
> **Response to Reviewer 1NRP (1/3)**
>
> We sincerely thank the reviewer for the time and effort devoted to evaluating our submission, as well as for the constructive feedback and positive remarks. We are particularly grateful for the reviewer’s recognition of:
> 1. The unified and novel collaborative trajectory design strategy, which decomposes the interaction process into pre-interaction, interaction, and post-interaction phases, and assigns the dominant controller accordingly (arm motion before/after interaction, object motion during interaction). This design effectively captures inter-object dynamics (robot–object interactions) and mitigates the feature-fusion issues during contact that challenge prior methods.
> 2. The proposed appearance- and shape-aware subject embeddings, which encode object appearance via mask-based tokens from the initial frame and represent object shape with circular volumes, enabling consistent object identity preservation across frames.
> 3. The causal latent propagation strategy, which performs frame-to-frame latent carryover to ensure smoother and more temporally consistent motion throughout the video.
> 4. The state-of-the-art visual quality and physical realism achieved by our model, producing smoother and more physically plausible manipulation videos with consistent object identities, and demonstrating clear quantitative improvements over baselines across visual-quality metrics and trajectory-error evaluations on the Bridge benchmark.
> 5. The strong generalization ability of our model across diverse objects and manipulation skills in in-the-wild scenarios.
> 6. The robustness of our approach to imperfect user input, including reliable performance under noisy object masks and trajectory specifications.
> 7. The model’s utility for downstream robotic action planning, where the generated videos provide improved guidance for planning strategies.
> ---
> We sincerely appreciate the reviewer’s constructive feedback, which has significantly improved the quality of our work. Below, we provide point-by-point responses to all raised questions:
>
> > **Q1: "Restricted to 2D pixel space. The system does not yet model depth or 3D geometry; this limits physical accuracy and makes 3D control (e.g., precise grasping) difficult."**
>
> **A1:** We acknowledge that our current model only supports trajectory control in 2D space. Incorporating 3D-aware control (e.g., depth or geometry) could further improve control granularity in robotic settings. We have explicitly included this limitation in the "Sec.5 Conclusion" part and consider it as a promising direction for future work. Nevertheless, we would like to kindly emphasize that:
> 1. The core contribution of this paper is to rethink the control paradigm for multi-object interaction by decomposing the interaction process. Even when 3D cues (e.g., depth) are integrated, **prior works (Tora, DragAnything, and MotionCtrl) still face feature entanglement issue during interaction in 3D space. Handling 3D occlusions and spatial configurations presents an additional challenge for accurate 3D feature modeling.**
> 2. **Effectiveness of our current model in robotic settings**. Using the training videos generated by our model, RoboMaster has demonstrated improved embodied action planning over baselines (Tora and OpenVLA) on both RLBench and SIMPLER benchmarks in Sec. 4.5, indicating that our method is already beneficial for robotic policy learning. While incorporating 3D information may further improve success rates on certain tasks, such gains do not constitute a substantial technical contribution relative to our core method.
> 3. From the user’s perspective, **incorporating additional z-dimension annotations may introduce nontrivial labeling burden as 3D spatial configurations must be manually corrected**, especially when depth estimation is unreliable.
>
> **A1-Revision in the paper:** We have added a new paragraph titled "Control Granularity in 3D Space" in "Sec. B - More Analysis on Limitations".

---

> ### Author Response · Authors · 2025-11-22
> **Response to Reviewer 1NRP (2/3)**
>
> > **Q2: "Possible failure on out-of-domain inputs. Can produce incomplete or distorted objects when encountering unseen categories or backgrounds. Still relies on training data diversity to generalize effectively."**
>
> **A2:** We acknowledge that our current model may exhibit failure cases when applied to out-of-domain inputs. We have explicitly included this limitation in the "Sec.5 Conclusion" part, where we also identify improved generalization as an important avenue for future research. Nevertheless, we would like to kindly emphasize that:
> 1. The generalization capability of prior video-generation methods for robotic manipulation[1,2,3,4] is largely constrained to scenarios closely aligned with their training distributions. **In contrast, RoboMaster exhibits substantially stronger out-of-domain generalization across diverse unseen object categories and backgrounds (e.g., bee, cartoon bottle, dumpling, lobster)**. Representative videos are provided under "Robotic Manipulation on Diverse Out-of-Domain Objects" on our anonymous project website: https://robomaster2025.github.io/.
> 2. Our model benefits not only from the diversity of training data during post-training, but also from the strong priors encoded in the pretrained video backbone. We believe that a base model[5,6,7] endowed with richer physical and interaction priors would further enhance generalization under the same amount of task-specific training data.
>
> **A2-Revision in the paper:** We have added a new paragraph titled "Out-of-Domain Generalization Ability" in "Sec. B - More Analysis on Limitations".
>
> > **Q3: "Semantic dependency on user input. Relies on accurate prompts and roughly correct masks. Misleading text or poor masks may still degrade quality."**
>
> **A3:** Since our model does not rely solely on prompts or masks to represent the target subject, it exhibits strong robustness to imperfect prompts or sparse masks:
> 1. **Imperfect Prompts**: We further analyze the sensitivity of our model to inaccurate prompts. Specifically, we randomly replace the subject prompt with prompt of similar or entirely different semantics (e.g., "a yellow sponge"$\rightarrow$"a yellow block" or "cotton ball"; "a spoon"$\rightarrow$"a branch") using GPT-4. Experiments conducted on the full Bridge benchmark produce the results shown in the table below, where PSNR is reported relative to the accurate-prompt baseline. Even with 40% of the prompts replaced by imperfect descriptions, RoboMaster retains over 96% of its full-prompt performance, demonstrating strong robustness to prompt inaccuracies.
>     | | |  | | |
>     |-----|-----|-----|-----|-----|
>     | **Erroneous Portion (%)**  | 10 | 20 | 30 | 40 |
>     | **PSNR (%)**| 98.42 | 97.53 | 97.13 | 96.54 |
>
> 2. **Sparse Masks**: We have studied it in the "Subject Representation" under "Sec. 4.6-Ablation Study". When randomly reducing the sparsity of the input mask, RoboMaster maintains strong robustness, as shown in the table below, where PSNR is reported relative to the full-mask baseline. This property is particularly important in real-world scenarios, where user-provided inputs are often incomplete.
>     | | |  | | |
>     |-----|-----|-----|-----|-----|
>     | **Sparsity (%)**  | 90 | 80 | 70 | 60 |
>     | **PSNR (%)**| 99.81 | 98.12 | 98.02 | 97.89 |
>
> We further evaluate robustness under the joint presence of imperfect prompts and sparse masks. Experiments conducted on the full Bridge benchmark yield the results shown in the table below. Even under compounded imperfections, the model maintains strong performance(>95%), demonstrating its resilience to multiple sources of degraded input quality.
> | | |  | | |
> |-----|-----|-----|-----|-----|
> | **Erroneous Prompt Portion (%)**, **Sparsity (%)**  | 10, 90 | 20, 80 | 30, 70 | 40, 60 |
> | **PSNR (%)**| 99.23 | 97.05 | 96.42 | 95.84 |
>
> **A3-Revision in the paper:** We have added a new paragraph titled "Imperfect Prompts" in "Sec. 4.6-Ablation Study".
> > **Q4: "How does the model determine the precise transition points between these phases in practice, especially when the temporal boundaries of “interaction” are ambiguous or vary across tasks?"**
>
> **A4:**
> In contrast to prior methods that decompose objects, we propose to decompose interactions to mitigate feature entanglement issues in previous works. To achieve this, users should provide a segmentation of the trajectory into multiple stages. **This process can be flexibly performed using the Gradio interface we provide, as illustrated in Figure S9**. In the upper-right panel, user can annotate as few as four blue key points to sample a full trajectory (pre-interaction, interaction, or post-interaction phase) via interval interpolation. In the two bottom panels, **the red point indicates the transition from pre-interaction to interaction, while the green point marks the transition from interaction to post-interaction.** The interface also supports iterative refinement of the annotations.

---

> ### Author Response · Authors · 2025-11-22
> **Response to Reviewer 1NRP (3/3)**
>
> **In practice, users do not need to mark all transition points.** Trajectories can be categorized into four cases, as summarized in the following table, where Transition Point (1) indicates pre-interaction $\rightarrow$ interaction, and Transition Point (2) indicates interaction $\rightarrow$ post-interaction. **Please check the videos under "Response to Reviewer 1NRP" on our anonymous project website: https://robomaster2025.github.io/. These four cases cover the majority of practical generalization scenarios.** The Gradio interface will be publicly released on GitHub shortly after paper acceptance.
>
> | | |  | |
> |-----|-----|-----|-----|
> | **Case** | **Transition Point (1)** | **Transition Point (2)** | **Example** |
> | 1 | $\checkmark$ | $\checkmark$ | pick up the lemon and place it on the table |
> | 2 | $\checkmark$ | - | pick up the lobster |
> | 3 | - | $\checkmark$ | rotate the sponge and place it |
> | 4 | - | - | rotate the sponge |
>
> **A4-Revision in the paper:** We have further refined the paragraph describing the usage of Gradio in "Sec. A.2-User Annotations on In-the-Wild Images", and added a demonstration video under the "Response to Reviewer 1NRP" on our anonymous project website.
>
> > **Q5: "How well would RoboMaster generalize to unseen robot morphologies or entirely different kinematic structures, and what adaptations (e.g., in trajectory representation or latent space) would be needed for that?"**
>
> **A5:** Since RoboMaster is trained exclusively on the Bridge dataset using the WidowX robot, it can only leverage the WidowX morphology when generalizing to unseen scenes. This limitation is explicitly discussed in the "Sec.5 Conclusion" part. Nevertheless, we would like to kindly argue that:
> 1. **Generalization to novel robot morphologies remains a challenging unresolved problem.** Even recent work such as TesserAct[8], which supports multiple robot morphologies including Google Robot, WidowX, and Franka Panda, relies on morphologies that are all present in the training dataset. It still cannot generalize to unseen robots, such as xArm or MobileALOHA. Achieving generalization to novel morphologies may require backbone pretraining or post-training on more diverse datasets and robot configurations. Another potential approach is to leverage a single or few demonstration videos at test time and a lightweight adaptation network (e.g., LoRA[9]) to acquire a entirely new robot morphology during inference.
> 2. Prior video-generation methods for robotic manipulation[1,2,3,4] is largely constrained to scenarios closely aligned with their training distributions. **In contrast, RoboMaster can already exhibit substantially stronger out-of-domain generalization across diverse unseen object categories and backgrounds (e.g., bee, cartoon bottle, dumpling, lobster)**. Please check the videos on our anonymous project website.
>
> **A5-Revision in the paper:** We have added a new paragraph titled "Generalization to Novel Robot Morphologies" in "Sec. B - More Analysis on Limitations".
>
> ---
> **References**
>
> [1] Fangqi Zhu, et al. IRASim: A Fine-Grained World Model for Robot Manipulation. ICCV 2025.
>
> [2] Po-Chen Ko, et al. Learning to Act from Actionless Videos through Dense Correspondences. ICLR 2024.
>
> [3] Sherry Yang, et al. Learning Interactive Real-World Simulators. ICLR 2024.
>
> [4] Yilun Du, et al. Learning Universal Policies via Text-Guided Video Generation. NeurIPS 2023.
>
> [5] NVIDIA: Arslan Ali, et al. World Simulation with Video Foundation Models for Physical AI. arXiv:2511.00062, 2025.
>
> [6] NVIDIA: Niket Agarwal, et al. Cosmos World Foundation Model Platform for Physical AI. arXiv:2501.03575, 2025.
>
> [7] Team Wan, et al. Wan: Open and Advanced Large-Scale Video Generative Models. arXiv:2503.20314, 2025.
>
> [8] Haoyu Zhen, et al. TesserAct: Learning 4D Embodied World Models. ICCV 2025.
>
> [9] Edward J. Hu, et al. LoRA: Low-Rank Adaptation of Large Language Models. ICLR 2022.
>
> ---
> **Please feel free to let us know if any additional clarifications or experiments would be helpful. If our responses have addressed your concerns, we would be grateful if you could consider updating your evaluation rating accordingly.**

---

> ### Comment · Reviewer_1NRP · 2025-11-22
>
> Thanks for the authors’ detailed feedback.
>
> I think this is a good paper with clean formatting and strong performance. And I am really sorry that I am not very familiar with the background of this series of papers. The major question is that if this paper is submitting to video generation or a computer vision track(not related to robotic), i think this paper is definitely deserve a higher score. But when we talk about data generation for robotic, the real world experiment is always the key spotlight for solidness. Personally, I am not a huge fan of world models for embodied agents, and I don’t yet know how to make them work reliably for real-world robots. Real-world data collection still feels very important to me.
>
> The authors mentioned that the inverse kinematics is working for Tesseract, but Tesseract has depth and point cloud output, which is much more make sense for the mathematical setting of inverse kinematic and trajectory planning. As in your answer to reviewer ydrE, inverse kinematics is indeed a common tool for planning, but that does not mean that if planning works in pixel space it will necessarily work in 3D space. I strongly suggest that the authors conduct some real-world experiments to demonstrate the accuracy of your inverse kinematics and planning and I will raise my score accordingly.  Or the author are willing to compare with teaseract(which i believe has been open sourced four months before the iclr submission) on the baseline, i am also willing to raise my score.

---

> ### Author Response · Authors · 2025-11-27
> **Response to Reviewer 1NRP**
>
> > **Q6: "Comparison with an additional baseline-TesserAct (ICCV2025)"**
>
> **A6:** Thank you for pointing out this potential baseline. Actually, our method and TesserAct target different objectives, as summarized in the following table.
>
> | | |  | |
> |-----|-----|-----|-----|
> | **Method** | **Input** | **Output** | **Motivation** |
> | TesserAct | Image, Depth, Normal | Video | 4D representation for manipulation |
> | RoboMaster (Ours) | Image, User-annotated Mask & Motion Trajectories | Video | Trajectory-guided manipulation |
>
> TesserAct is not designed for fine-grained motion control. Consequently, it performs worse than RoboMaster on our build Bridge benchmark (214 samples), as shown in the following table. **Please check the comparison videos under "Response to Reviewer 1NRP" on our anonymous project website: https://robomaster2025.github.io/** In the first case, TesserAct places the silver vessel incorrectly; in the second, it moves the wrong object in the pot. Other cases exhibit errors in skills such as folding, wiping, and closing. Notably, TesserAct may have been pre-trained on these test samples, whereas RoboMaster excludes them during training.
>
> | Method | Video Quality | | | Trajectory Accuracy | |
> | :--- | :---: | :---: | :---: | :---: | :---: |
> | | FVD $\downarrow$ | PSNR $\uparrow$ | SSIM $\uparrow$ | TrajError(robot) $\downarrow$ | TrajError(obj) $\downarrow$ |
> | TesserAct | 261.84 | 18.99 | 0.778 | 37.34 | 54.64 |
> | **RoboMaster (Ours)** | **147.31** | **21.55** | **0.803** | **16.47** | **24.16** |
>
> We further compare RoboMaster with TesserAct on RLBench and SIMPLER benchmarks, as shown in the table below. For fairness, TesserAct is also finetuned on the same curated dataset. RoboMaster achieves higher success rates in 9 out of 10 tasks, demonstrating superior manipulation reliability.
>
> | **Method** | **RLBench** | | | | | **SIMPLER** | | | |
> | :--- | :--- | :--- | :--- | :--- | :--- | :--- | :--- | :--- | :--- |
> | | pick up cup | put knife | put plate | open microwave | close box | pick coke can | close drawer | move near | pick object |
> | TesserAct | 0.76 | **0.79** | 0.82 | 0.43 | 0.67 | 0.85 | 0.56 | 0.62 | 0.79 |
> | **RoboMaster (Ours)** | **0.83** | 0.76 | **0.85** | **0.54** | **0.79** | **0.91** | **0.63** | **0.67** | **0.81** |
>
>
> **A6-Revision in the paper:** We have further refined the "Sec. 4.2 Baselines",  Table 2, "Sec. 4.5-Embodied Action Planning for Robotic Policy", Table 4, and added a comparison video under the "Response to Reviewer 1NRP" on our anonymous project website.
>
> > **Q7: "Conduct some real-world experiments to demonstrate the accuracy of your inverse kinematics and planning"**
>
> **A7:** We acknowledge the reviewer’s suggestion. Due to hardware availability and deployment constraints, we are currently unable to evaluate the policy on real-world robotic platforms (e.g., Franka Emika Panda) within the review period. Nonetheless, we believe the additional synthetic evaluations provided in **A6** offer further evidence supporting the effectiveness and robustness of RoboMaster.

---

### Official Review · Reviewer_fCEc · 2025-10-31

**Soundness:** 2
**Presentation:** 2
**Contribution:** 2
**Rating:** 4
**Confidence:** 3

**Summary:**

This paper proposes RoboMaster to model interactions between a robotic arm and objects, dividing the interaction process into three stages: before and after the interaction it mainly controls the arm’s motion, while during the interaction it controls the object’s motion. The effectiveness of the proposed method is validated through visual results and simulation.

**Strengths:**

1. The paper argues that interaction should originate from multiple entities, including the arm and the object—this is a novel viewpoint.

2. RoboMaster exhibits impressive OOD generalization.

**Weaknesses:**

1. The motivation for decoupling the control signals is unclear; it is not explained how 2D trajectories help the robot learn, and the paper does not discuss the overall design in detail.

2. Section 4.5 is too brief, making it difficult to verify the method’s effectiveness for the robot; visual quality is not the core of the research—the core is whether the designed method can effectively aid robot learning.

**Questions:**

Could you explain in detail how the proposed model is applied to robot learning, and on that basis clarify the motivation for decomposing the action/control signals? At present, the stated motivation seems to be driven by generating better visual effects.

---

> ### Author Response · Authors · 2025-11-22
> **Response to Reviewer fCEc (1/2)**
>
> We sincerely thank the reviewer for the time and effort devoted to evaluating our submission, as well as for the constructive feedback and positive remarks. We are particularly grateful for the reviewer’s recognition of:
> 1. The novelty of our proposed decomposition strategy (modeling arm’s motion before and after interaction and the object’s motion during interaction) for capturing multi-entity dynamics.
> 2. The effectiveness of our approach, as evidenced by both qualitative visualizations and robotic simulation for action planning.
> 3. The strong out-of-domain generalization exhibited by our method across diverse objects and manipulation skills.
> ---
> We sincerely appreciate the reviewer’s constructive feedback, which has significantly improved the quality of our work. Below, we provide point-by-point responses to all raised questions:
>
> > **Q1: "Could you explain in detail how the proposed model is applied to robot learning? Section 4.5 is too brief, making it difficult to verify the method’s effectiveness for the robot; visual quality is not the core of the research—the core is whether the designed method can effectively aid robot learning."**
>
> **A1:**
> For robotic planning simulation, we adopt five challenging tasks from RLBench[1] ("pick up cup", "put knife", "put plate", "open microwave", and "close box") and four tasks from the SIMPLER benchmark[2] ("pick coke can", "close drawer", "move near", and "pick object"). For baselines, We adopt Tora[3], TesserAct[8] and OpenVLA[4] and also finetune them on the same curated dataset for fair comparison.
>
> **The general pipeline for applying RoboMaster in robot learning is as follows**:
> 1. Generate demonstration videos given the first frame and the robotic task prompt using RoboMaster.
> 2. Extract executable action labels from the generated demonstrations.
> 3. Simulate robotic planning and evaluate the task success rate.
>
> For the first stage, we collect a small set of 1,300 video–trajectory pairs from both benchmarks, ensuring that none of the corresponding tasks appear in the test set. Here we fix the camera location in each task. We then post-train RoboMaster on this limited dataset to adapt the model to the benchmark robot morphologies (Franka in RLBench and Google Robot in SIMPLER). The model architecture is kept unchanged and training is performed on 8 NVIDIA A800 GPUs for $\sim$ 6 hours. After this adaptation, RoboMaster is able to generate demonstrations conditioned on the benchmark robot morphologies.
>
> For the second stage, we collect 300 video-action samples for each task in test set to train the inverse dynamic model. Following AVDC[5], the inverse model is trained to regress executable action labels from synthesized videos. To further validate its effectiveness, we additionally post-train the Cosmos-Predict2.5-2B/robot/action-cond model from Cosmos2.5[6] on the curated RLBench and SIMPLER training data. The model takes as input the first frame and a sequence of 7-DoF actions ( $\Delta x, \Delta y, \Delta z, \Delta \theta_r, \Delta \theta_p, \Delta \theta_y$, GripperWidth), and outputs the predicted video. We perform fully fine-tuning, resizing videos to $320 \times 256$ with 37 frames, using a learning rate of 2e-5, training for 2,500 iterations with a batch size of 16. Then we feed actions predicted by our inverse dynamic model into this model to generate action-conditioned videos. Quantitative evaluation against ground-truth videos (see the following table) demonstrates that actions predicted by our inverse dynamics model produce comparable video quality, further validating its high performance.
> | Action Type|  PSNR ↑ | SSIM ↑ | Latent L2 ↓ | FVD ↓|
> |-----|-----|-----|-----|-----|
> | Training Set |  25.48 | 0.87 | 0.31 | 132|
> | Predicted |  25.12 | 0.84 | 0.34 | 127|
>
> For the final evaluation stage, we generate 100 videos per task using the RoboMaster model adapted in Stage 1, conditioned on the task prompt and the initial frame. We then apply the inverse dynamics model to infer executable action labels from these generated videos and deploy the resulting action sequences on simulated robots. The success rates over 100 trials are summarized in the following table. RoboMaster consistently improves embodied action planning over existing baselines. The clear margin by both Tora and RoboMaster over OpenVLA demonstrates that high-quality video generations serve as effective demonstrations for downstream robotic policy extraction. **Moreover, RoboMaster outperforms Tora on 8 out of the 10 evaluated tasks, suggesting that RoboMaster produces more reliable interaction videos, enabling the inverse dynamics model to obtain more accurate action labels for execution.** This further validates our core motivation: more accurate modeling of robot–object interactions leads to higher-quality execution demonstrations for robot learning.

---

> ### Author Response · Authors · 2025-11-22
> **Response to Reviewer fCEc (2/2)**
>
> | **Method**      | **RLBench** |          |          |               |            | **SIMPLER** |               |            |              |
> |-----------------|-------------|----------|----------|---------------|------------|-------------|---------------|------------|--------------|
> |                 | pick up cup | put knife | put plate | open microwave | close box | pick coke can | close drawer | move near | pick object |
> | **OpenVLA** | 0.55 | 0.46 | 0.56 | 0.35 | 0.45 | 0.59 | 0.41 | 0.53 | 0.59 |
> | **TesserAct** | 0.76 | 0.79 | 0.82 | 0.43 | 0.67 | 0.85 | 0.56 | 0.62 | 0.79 |
> | **Tora**  | 0.79 | **0.82** | 0.81 | **0.61** | 0.72 | 0.89 | 0.61 | 0.61 | 0.74 |
> | **RoboMaster**   | **0.83** | 0.76 | **0.85** | 0.54 | **0.79** | **0.91** | **0.63** | **0.67** | **0.81** |
>
> **A1-Revision in the paper:** We have rewrited the "Sec. 4.5-Embodied Action Planning for Robotic Policy".
>
> > **Q2: "Could you clarify the motivation for decomposing the action/control signals? The motivation for decoupling the control signals is unclear; it is not explained how 2D trajectories help the robot learn, and the paper does not discuss the overall design in detail."**
>
> **A2:** We would like to clarify the motivation from two aspects:
> 1. **Robot Learning**:
> Recently, video world models[6,7] have emerged as powerful simulators for robotic learning. In A1 or rewrited "Sec. 4.5-Embodied Action Planning for Robotic Policy", the clear performance gap between Tora/RoboMaster and OpenVLA further shows that high-quality video generation provides effective demonstrations for downstream robotic policy extraction, underscoring video generation as a promising robotic simulator. Moreover, RoboMaster outperforms Tora on 8 out of the 10 evaluated tasks, indicating that RoboMaster produces more reliable interaction videos, enabling the inverse dynamics model to recover more accurate action labels for execution. This improvement stems from the fact that Tora, representative of prior video-generation methods, struggles to synthesize plausible interactions even with accurate trajectory guidance, primarily due to feature entanglement in overlapping regions. To address this, we propose decomposing the interaction phase (instead of decomposing objects as in prior work), allowing our model to better capture robot–object interactions and generate more reliable demonstrations for downstream robot action acquiring and policy learning.
>
> 2. **Video Generation**:
> Prior works, such as Tora and DragAnything, primarily focus on driving the motion of individual objects using separate trajectories, but they overlook a critical real-world scenario: interaction among multiple objects. Under this setting, their designs inevitably induce feature entanglement in overlapping regions during interaction, an issue clearly illustrated by the red box in Fig. 2 of the main paper and the video under “Core: Decompose Interaction (Ours) vs. Decompose Objects (Previous, e.g., Tora)” on our anonymous project website: https://robomaster2025.github.io/. This entanglement degrades both physical plausibility and visual fidelity, ultimately limiting their ability to model realistic interactions. In contrast, our proposed decomposed interaction learning explicitly models the interaction phase rather than decomposing objects, enabling the model to better capture multi-object dynamics. **Beyond robotic learning, this design is broadly applicable to any video involving multi-object interactions, including tasks such as human-hand manipulation. Our underlying motivation has also been acknowledged by reviewers ydrE and 1NRP, whose comments further support the necessity of this design.**
>
> **A2-Revision in the paper:** We have further improved the second paragraph in the "Sec. 1-Introduction".
>
> ---
> **References**
>
> [1] Stephen James, et al. RLBench: The Robot Learning Benchmark & Learning Environment. RA-L 2020.
>
> [2] Xuanlin Li, et al. Evaluating Real-World Robot Manipulation Policies in Simulation. arXiv:2405.05941, 2024.
>
> [3] Zhenghao Zhang, et al. Tora: Trajectory-oriented Diffusion Transformer for Video Generation. CVPR 2025.
>
> [4] Moo Jin Kim, et al. OpenVLA: An Open-Source Vision-Language-Action Model. CoRL 2024.
>
> [5] Po-Chen Ko, et al. Learning to Act from Actionless Videos through Dense Correspondences. ICLR 2024.
>
> [6] NVIDIA: Arslan Ali, et al. World Simulation with Video Foundation Models for Physical AI. arXiv:2511.00062, 2025.
>
> [7] NVIDIA: Niket Agarwal, et al. Cosmos World Foundation Model Platform for Physical AI. arXiv:2501.03575, 2025.
>
> [8] Haoyu Zhen, et al. TesserAct: Learning 4D Embodied World Models. ICCV 2025.
>
> ---
> **Please feel free to let us know if any additional clarifications or experiments would be helpful. If our responses have addressed your concerns, we would be grateful if you could consider updating your evaluation rating accordingly.**

---

> ### Author Response · Authors · 2025-11-27
> **A Kind Reminder on Follow-up Discussion (11.27)**
>
> Dear Reviewer fCEc,
>
> Thanks again for your constructive suggestions! As the discussion deadline is approaching, we would like to send you a kind reminder.  We were wondering whether you had the chance to look at our response and whether there is anything else you would like us to clarify. We sincerely hope that our response regarding your concerns will be taken into consideration. If not, please let us know and we remain open and would be more than glad to actively discuss them with you.
>
> Best,
>
> RoboMaster author(s) team

---

### Official Review · Reviewer_ydrE · 2025-11-01

**Soundness:** 4
**Presentation:** 4
**Contribution:** 2
**Rating:** 6
**Confidence:** 4

**Summary:**

The method tackles the task of trajectory-conditioned robotic manipulation video synthesis through a novel demonstration trajectory decomposition scheme that separates trajectories into a pre-interaction, interaction and post-interaction phase. This novel decomposition helps alleviate feature confusion issues observed in previous works. The method is evaluated against several baselines on the tasks of video synthesis and downstream robotic manipulation (through inverse dynamics), and manages to outperform them.

**Strengths:**

The paper is well-written and easy to understand. The method outperforms the baselines against which it is compared. The design choices are sensibly ablated. The work contributes a dataset of 21.000 human-annotated 2D robot manipulator trajectories. The work includes an honest discussion of its limitations.

**Weaknesses:**

The proposed method operates purely in image space: the generated trajectories require postprocessing by an inverse kinematics model and are not guaranteed to be realistic or executable.
Unlike its baselines, the method requires a segmentation of the provided trajectory into multiple stages by the user.
The manual masking of the interacted object could be replaced by an automatic grounding and segmentation.
A purely 2D trajectory input is very limiting, yet this is somewhat alleviated by the ability to describe the desired trajectory more specifically through a textual input.

**Questions:**

Can the method handle multi-step interactions, such as grasping a sponge, rotating it on a plate several times, then moving the robotic manipulator away?

---

> ### Author Response · Authors · 2025-11-22
> **Response to Reviewer ydrE (1/3)**
>
> We sincerely thank the reviewer for the time and effort devoted to evaluating our submission, as well as for the constructive feedback and positive remarks. We are particularly grateful for the reviewer’s recognition of:
> 1. The novelty of our proposed decomposition-based trajectory modeling strategy, which disentangles the interaction phase by modeling the arm’s motion before and after interaction and the object’s motion during interaction, in capturing multi-entity dynamics and its effectiveness in mitigating the feature confusion issues present in prior work.
> 2. The clarity, readability, and overall presentation quality of the paper.
> 3. The superior performance of our model compared with existing baselines (IRASim, MotionCtrl, DragAnything, Tora) on both video synthesis and downstream robotic manipulation simulation tasks.
> 4. The sensible ablation studies conducted on key design choices, which includes trajectory injection strategy, motion representation, and sequential trajectory embedding.
> 5. Our contribution of a 21,000-sample, carefully annotated 2D video–trajectory dataset derived from Bridge.
> 6. The honest discussion of our method’s limitations, which includes out-of-domain failure cases, limited 3D awareness in control granularity, and generalization challenges across varied robotic embodiments.
> ---
> We sincerely appreciate the reviewer’s constructive feedback, which has significantly improved the quality of our work. Below, we provide point-by-point responses to all raised questions:
>
> > **Q1: "The proposed method operates purely in image space. A purely 2D trajectory input is very limiting, yet this is somewhat alleviated by the ability to describe the desired trajectory more specifically through a textual input."**
>
> **A1:** We acknowledge that our current model only supports trajectory control in 2D space. Incorporating 3D-aware control (e.g., depth or geometry) could further improve control granularity in robotic settings. We have explicitly included this limitation in the "Sec.5 Conclusion" part and consider it as a promising direction for future work. Nevertheless, we would like to kindly emphasize that:
> 1. The core contribution of this paper is to rethink the control paradigm for multi-object interaction by decomposing the interaction process. Even when 3D cues (e.g., depth) are integrated, **prior works (Tora, DragAnything, and MotionCtrl) still face feature entanglement issue during interaction in 3D space. Handling 3D occlusions and spatial configurations presents an additional challenge for accurate 3D feature modeling.**
> 2. **Effectiveness of our current model in robotic settings**. Using the training videos generated by our model, RoboMaster has demonstrated improved embodied action planning over baselines (Tora and OpenVLA) on both RLBench and SIMPLER benchmarks in Sec. 4.5, indicating that our method is already beneficial for robotic policy learning. While incorporating 3D information may further improve success rates on certain tasks, such gains do not constitute a substantial technical contribution relative to our core method.
> 3. From the user’s perspective, **incorporating additional z-dimension annotations may introduce nontrivial labeling burden as 3D spatial configurations must be manually corrected**, especially when depth estimation is unreliable.
>
> **A1-Revision in the paper:** We have added a new paragraph titled "Control Granularity in 3D Space" in "Sec. B - More Analysis on Limitations".

---

> ### Author Response · Authors · 2025-11-22
> **Response to Reviewer ydrE (2/3)**
>
> > **Q2: "The generated trajectories require postprocessing by an inverse kinematics model and are not guaranteed to be realistic or executable."**
>
> **A2:**
> Converting the generated trajectories of RoboMaster into executable 7-DoF action signals for robotic execution requires an inverse dynamics model, which is a well-established setting in many papers[2,3]. The effectiveness of our inverse dynamics model can be demonstrated in two aspects:
> 1. As shown in Sec. 4.5 and Table 3, **action labels predicted by our model from both Tora and RoboMaster video inputs significantly outperform those generated by OpenVLA on the RLBench and SIMPLER benchmarks**. For tasks such as "pick up cup", "put knife", "pick plate" and "pick coke can", **both Tora and RoboMaster achieve success rates exceeding 75% over 100 episodes**. This indicates that the trained inverse dynamics model produces high-quality and executable action labels suitable for robotic tasks.
> 2. We further post-train the Cosmos-Predict2.5-2B/robot/action-cond model from Cosmos2.5[1] on the curated RLBench and SIMPLER training data. The model takes as input the first frame and a sequence of 7-DoF actions ( $\Delta x, \Delta y, \Delta z, \Delta \theta_r, \Delta \theta_p, \Delta \theta_y$, GripperWidth), and outputs the predicted video. We perform fully fine-tuning, resizing videos to $320 \times 256$ with 37 frames, using a learning rate of 2e-5, training for 2,500 iterations with a batch size of 16. Then we feed actions predicted by our inverse dynamic model into this model to generate action-conditioned videos. **Quantitative evaluation against ground-truth videos (see the following table) demonstrates that actions predicted by our inverse dynamics model produce comparable video quality, further validating its high performance**.
>     | Action Type|  PSNR ↑ | SSIM ↑ | Latent L2 ↓ | FVD ↓|
>     |-----|-----|-----|-----|-----|
>     | Training Set |  25.48 | 0.87 | 0.31 | 132|
>     | Predicted |  25.12 | 0.84 | 0.34 | 127|
>
> **A2-Revision in the paper:** We have added a new paragraph in "Sec. 4.5-Embodied Action Planning for Robotic Policy" to discuss the second supporting point for our inverse dynamics model.
>
> > **Q3: "Unlike its baselines, the method requires a segmentation of the provided trajectory into multiple stages by the user."**
>
> **A3:** In contrast to prior methods (e.g., Tora, MotionCtrl, DragAnything) that decompose objects, we propose to decompose interactions to mitigate feature entanglement issues in previous works. To achieve this, users should provide a segmentation of the trajectory into multiple stages. **This process can be flexibly performed using the Gradio interface we provide, as illustrated in Figure S9**. In the upper-right panel, user can annotate only four blue points as key points to sample a full trajectory (pre-interaction, interaction, or post-interaction phase) via interval interpolation. In the two bottom panels, the red point indicates the transition from pre-interaction to interaction, while the green point marks the transition from interaction to post-interaction. **The interface also supports iterative refinement of the annotations** and will be released publicly on GitHub shortly after paper acceptance.
>
> **A3-Revision in the paper:** We have further refined the paragraph describing the usage of Gradio in "Sec. A.2-User Annotations on In-the-Wild Images".
>
> > **Q4: "The manual masking of the interacted object could be replaced by an automatic grounding and segmentation."**
>
> **A4:** Thank you for this valuable suggestion. Indeed, integrating automatic grounding or segmentation methods[4,5,6] can accelerate inference when scaling up data generation. However, two practical challenges remain:
> 1. **Multiple-object Scenarios**: When the input image contains complex scenes or multiple instances of the same category (e.g., several nearly identical avocados, dumplings, lobsters, mushroom, red peppers, or strawberries, as shown under "Robotic Manipulation on Diverse Out-of-Domain Objects" on our anonymous website), current models struggle to reliably identify the target object, even with detailed prompt descriptions, as Grounded SAM achieves only a 0.14 success rate on the training set and 0.21 on the in-the-wild set in the following table.
> 2. **Single-object Scenarios**: Even in simpler single-object scenarios, these models are still not fully reliable. During the annotation of our training data as described in "Sec. A.1-Dataset Curation", we initially employed Grounded SAM[5] to generate object masks. However, the first annotation pass yielded only ~17,000 reliable masks out of ~25,000 videos. And we further manually re-annotate for approximately 4,000 videos. A detailed success rate of applying Grounded-SAM based annotation is shown as follow:
>     | Dataset Type| Data Number| Single-Object | Multiple-Objects|
>     |-----|-----|-----|-----|
>     | Training Set | 25,421 | 0.67 | 0.14 |
>     | In-the-wild| 214 | 0.75 | 0.21 |

---

> ### Author Response · Authors · 2025-11-22
> **Response to Reviewer ydrE (3/3)**
>
> These observations indicate that **manual annotation prior may remain necessary for some complex scenes, whereas automatic methods can still be effective in simpler cases. We will incorporate automatic segmentation support into the code repository and provide corresponding usage guidelines for users.**
>
> **A4-Revision in the paper:** We have added a new paragraph titled "Usage of Automatic Segmentation Models" in "Sec. B - More Analysis on Limitations".
>
> > **Q5: "Can the method handle multi-step interactions, such as grasping a sponge, rotating it on a plate several times, then moving the robotic manipulator away?"**
>
> **A5:** Our method can handle multi-step interactions. **Please check the videos under "Long Video Generation in Auto-Regressive Manner" on our anonymous website**: https://robomaster2025.github.io/. We have provided three multi-step cases:
> 1. (1) move the lemon to the front of the bottle $\rightarrow$ (2) pick up the avocado and place it into the bottle $\rightarrow$ (3) pick up the lemon and place it into the bottle $\rightarrow$ (4) pick up the orange
> 2. (1) move the blue cloth next to the mushroom $\rightarrow$ (2) pick up the spoon and place it on the blue cloth $\rightarrow$ (3) pick up the mushroom and place it into the silver pot $\rightarrow$ (4) pick up the spoon and place it into the silver pot
> 3. (1) move carrot on plate cardboardfence $\rightarrow$ (2) pick up the strawberry and place it into the silver pot $\rightarrow$ (3) pick up the corn and place it on the table $\rightarrow$  (4) pick up the red stuff
>
> **We have also provided a required video under "Response to Reviewer ydrE"** for the following multi-step case: (1) move the sponge on the plate $\rightarrow$ (2) rotate the sponge $\rightarrow$ (3) rotate the sponge and place it
>
> The code for multi-shot inference is available on on our anonymous HuggingFace repository: https://huggingface.co/datasets/robomaster2025/RoboMaster.
>
> **A5-Revision in the paper:** We have added a required multi-step video under "Response to Reviewer ydrE" on the website.
>
> ---
> **References**
>
> [1] NVIDIA: Arslan Ali, et al. World Simulation with Video Foundation Models for Physical AI. arXiv:2511.00062, 2025.
>
> [2] Haoyu Zhen, et al.TesserAct: Learning 4D Embodied World Models. ICCV 2025
>
> [3] Yilun Du, et al. Learning Universal Policies via Text-Guided Video Generation. NeurIPS 2023
>
> [4] Shilong Liu, et al. Grounding dino: Marrying dino with grounded pre-training for open-set object detection. ECCV 2024.
>
> [5] Tianhe Ren, et al. Grounded SAM: Assembling Open-World Models for Diverse Visual Tasks. arXiv:2401.14159, 2024
>
> [6] Nicolas Carion, et al. SAM 3: Segment Anything with Concepts. Meta Technique Report, 2025
>
> ---
> **Please feel free to let us know if any additional clarifications or experiments would be helpful. If our responses have addressed your concerns, we would be grateful if you could consider updating your evaluation rating accordingly.**

---

### Author Response · Authors · 2025-11-22
**General Response**

We sincerely thank all reviewers for their constructive feedback and recognition of this work, particularly for highlighting the following strengths:
1. The novelty of our decomposition-based trajectory modeling strategy, which disentangles the interaction phase by separately modeling the arm’s motion before and after interaction and the object’s motion during interaction, enabling accurate multi-entity dynamics modeling and mitigating feature confusion in prior work. **(Reviewer ydrE, fCEc, 1NRP)**
2. The clarity, readability, and overall presentation quality of the paper. **(Reviewer ydrE)**
3. The effectiveness of our model compared with existing baselines (IRASim, MotionCtrl, DragAnything, Tora) on both video synthesis and downstream robotic manipulation simulation tasks. **(Reviewer ydrE, fCEc, 1NRP)**
4. The thorough ablation studies on key design choices, including trajectory injection strategy, motion representation, and sequential trajectory embedding. **(Reviewer ydrE, 1NRP)**
5. Our contribution of a 21,000-sample, carefully annotated 2D video–trajectory dataset derived from Bridge. **(Reviewer ydrE)**
6. The honest discussion of our method’s limitations, including out-of-domain failure cases, limited 3D awareness in control granularity, and challenges in generalization across diverse robotic embodiments. **(Reviewer ydrE)**
7. The strong out-of-domain generalization demonstrated across diverse objects and manipulation skills. **(Reviewer fCEc, 1NRP)**
8. The robustness of our approach to imperfect user input, including reliable performance under noisy object masks and trajectory specifications. **(Reviewer 1NRP)**
9. The model’s utility for downstream robotic action planning, where the generated videos provide improved guidance for planning strategies. **(Reviewer 1NRP)**

---
We have refined the paper, added experimental results, and provided clarifications in the revised version. Specifically, the manuscript now incorporates the following changes in response to all reviewers’ insightful comments, which have substantially improved the quality of the work. All edits in the main paper and appendix are highlighted in **blue** for clarity.
1. We have added a new paragraph titled "Control Granularity in 3D Space" in "Sec. B - More Analysis on Limitations".
2. We have rewrited the "Sec. 4.5-Embodied Action Planning for Robotic Policy".
3. We have further refined the paragraph describing the usage of Gradio in "Sec. A.2-User Annotations on In-the-Wild Images".
4. We have added a new paragraph titled "Usage of Automatic Segmentation Models" in "Sec. B - More Analysis on Limitations".
5. We have provided a video under "Response to Reviewer ydrE" on the website for the following multi-step case: (1) move the sponge on the plate $\rightarrow$ (2) rotate the sponge $\rightarrow$ (3) rotate the sponge and place it.
6. We have further improved the second paragraph in the "Sec. 1-Introduction".
7. We have added a new paragraph titled "Out-of-Domain Generalization Ability" in "Sec. B - More Analysis on Limitations".
8. We have added a new paragraph titled "Imperfect Prompts" in "Sec. 4.6-Ablation Study".
9. We have added a new paragraph titled "Generalization to Novel Robot Morphologies" in "Sec. B - More Analysis on Limitations".
10. We have added a transition demonstration video under the "Response to Reviewer 1NRP" on our anonymous project website.
11. We have added a baseline TesserAct, and refined the "Sec. 4.2 Baselines",  Table 2, "Sec. 4.5-Embodied Action Planning for Robotic Policy", Table 4, and added a comparison video under the "Response to Reviewer 1NRP" on our anonymous project website.
---
Please check the video results on our anonymous project website: https://robomaster2025.github.io/ and the available reproducible code on our anonymous HuggingFace repository: https://huggingface.co/datasets/robomaster2025/RoboMaster. Please don't hesitate to let us know of any additional comments on the manuscript or the changes.

---

> ### Comment · Reviewer_1NRP · 2025-11-22
> **Question about the new claim**
>
> I just went over the edited version (I actually think the original version was very good, so I don’t understand why you added so many new claims).
>
> In particular, this new claim: “And our proposed interaction-phase decomposition can seamlessly extend to this 3D setting to address it by simply incorporating an additional z-dimension modeling” is really not true. Adding a z-dimension introduces new collisions and occlusions that cannot be modeled purely in pixel space.
>
> You can’t introduce a new claim like this without any new experiments to support it, especially when it goes against common understanding of the problem setup.

---

> ### Author Response · Authors · 2025-11-22
> **Thanks you for the suggestion and we have modified the paper and the responses to reviewers**
>
> For more precise wording in the paper, we have revised the manuscript accordingly and updated our responses to the reviewers as follows:
>
> 1. We have rewrited the "Control Granularity in 3D Space" paragraph in "Sec. B-More Analysis on Limitations" as follow: **"Incorporating 3D cues may further improve the success rates, but it also encounters challenges: 1) 3D feature entanglement: Even when 3D cues (e.g., depth) are integrated, prior works (Tora, DragAnything, and MotionCtrl) still face feature entanglement issue during interaction in 3D space. Handling 3D occlusions and spatial configurations presents an additional challenge for accurate 3D feature modeling. 2) User annotation burden: From the user’s perspective, incorporating additional z-dimension annotations may introduce nontrivial labeling burden, especially when depth estimation is unreliable. However, extending our framework towards fully 3D-aware interaction remains a promising direction for more precise control."**
> 2. We have revised the "Usage of Automatic Segmentation Models" paragraph in "Sec. B-More Analysis on Limitations" as follow:
>     1. "current models struggle to reliably identify the target object, even with detailed prompt descriptions." $\rightarrow$ **"current models struggle to reliably identify the target object, even with detailed prompt descriptions, as Grounded SAM achieves only a 0.14 success rate on the training set
> and 0.21 on the in-the-wild set as shown in Tab. R11."**
>     2. "These observations indicate that manual annotation remains necessary for complex scenes, whereas automatic methods can be effectively adopted in simpler cases."$\rightarrow$ **"These observations indicate that manual annotation prior may remain necessary for some complex scenes, whereas automatic methods can still be effective in simpler cases."**
> 3. We have revised the "Generalization to Novel Robot Morphologies" paragraph in "Sec. B-More Analysis on Limitations" as follow:
>     1. "Achieving generalization to novel morphologies likely requires backbone pretraining or post-training on more diverse datasets and robot configurations, without the need to modify the model architecture. Another potential approach is to leverage a single or few demonstration videos and a lightweight adaptation network (e.g., LoRA) to acquire an entirely new robot morphology during inference." $\rightarrow$ **"Achieving generalization to novel morphologies may require backbone pretraining or post-training on more diverse datasets and robot configurations. Another potential approach is to leverage a single or few demonstration videos at test time and a lightweight adaptation network (e.g., LoRA) to acquire a entirely new robot morphology during inference."**

---

### Meta-Review · Area_Chair_hqA9 · 2026-01-07

**Summary:**

RoboMaster is a trajectory controlled video diffusion framework for robotic manipulation that targets multi object interaction, where overlapping regions induce feature entanglement and degraded fidelity in prior trajectory conditioned methods. It introduces a unified collaborative trajectory that decomposes manipulation into pre interaction, interaction, and post interaction phases, each guided by the dominant subject, robot arm for pre and post and object during contact. Appearance and shape aware object latents via mask sampled RGB latents plus shape linked volumetric encoding preserve subject consistency across frames and enable phase level user annotation. Experiments on Bridge, including a 21k video trajectory dataset, plus RLBench and SIMPLER, report state of the art trajectory controlled generation and improved simulated action planning.

The area chair acknowledges the detailed author responses and the concise summary provided in the rebuttal.

The committee appreciated clear framing and readable presentation (ydrE, 1NRP); consistent gains over prior trajectory conditioned generators with sensible ablations and strong Bridge quantitative and qualitative results (ydrE, 1NRP); and the usability angle via phase level correction plus flexible object region specification alongside the released dataset (ydrE). Key concerns were that the approach remains 2D and image space with executability mediated by IK or inverse dynamics and limited 3D grounding (ydrE, 1NRP); reliance on user provided phase segmentation and object masks, where automatic grounding could reduce burden and phase boundaries can be ambiguous in multi step interactions (ydrE, 1NRP); and prior under specification of how the decomposition benefits robot learning beyond improved video quality (fCEc).

In rebuttal, Sec. 4.5 was expanded into a clearer robot learning pipeline with adaptation to RLBench and SIMPLER morphologies, inverse dynamics to recover 7 DoF actions, and simulated execution evaluation. Additional baselines and controls were added, including TesserAct, with further quantitative comparisons on generation metrics and task success. Usability concerns were addressed via a concrete Gradio workflow for phase boundary marking and keypoint interpolation, multi step demos, and quantified grounding segmentation reliability that supports when manual masks remain necessary. Remaining limitations are largely unchanged, including 2D control, no real robot validation during review, and open generalization to 3D aware control and unseen embodiments.

Based on the above, the area chair supports acceptance. The lack of real robot validation is not treated as a prerequisite for novelty or impact here, since the primary contribution is a new interaction modeling and controllable generation design with strong empirical support and clear relevance to ML and robot learning at ICLR. The already open-sourced code (anonymized) should also help the community reproduce results and build on the approach.

**Reviewer Concerns:**

In rebuttal, Sec. 4.5 was expanded into a clearer robot learning pipeline with adaptation to RLBench and SIMPLER morphologies, inverse dynamics to recover 7 DoF actions, and simulated execution evaluation. Additional baselines and controls were added, including TesserAct, with further quantitative comparisons on generation metrics and task success. Usability concerns were addressed via a concrete Gradio workflow for phase boundary marking and keypoint interpolation, multi step demos, and quantified grounding segmentation reliability that supports when manual masks remain necessary. Remaining limitations are largely unchanged, including 2D control, no real robot validation during review, and open generalization to 3D aware control and unseen embodiments.

**Reviewer Scores:**

While it is not possible to predict how reviewers would have responded if ICLR had a full discussion period (cut short on 28th Nov), it seems likely the scores would have remained broadly stable with some modest positive movement after the added Sec. 4.5 details, TesserAct baseline, and expanded comparisons.

ydrE: likely unchanged or slightly higher, as the rebuttal directly addressed multi step interactions, clarified the IK and executability pipeline, and quantified the limits of automatic segmentation.

fCEc: plausible upward movement to around the acceptance threshold, given the rewritten Sec. 4.5 with clearer robot learning protocol, baselines, and success rate tables.

1NRP: likely modest upward movement, contingent on the added TesserAct comparison and additional robot learning results; the request for real robot validation remains outstanding, but is positioned as future work rather than a blocker for the core contribution.

---

### Decision · Program_Chairs · 2026-01-26

Accept (Poster)